# Meta-World+: An Improved, Standardized, RL Benchmark

**Reginald McLean**[*]
Toronto Metropolitan University
Farama Foundation
reginald.mclean@torontomu.ca

**Evangelos Chatzaroulas** [*]
University of Surrey
e.chatzaroulas@surrey.ac.uk

**Luc McCutcheon**
University of Surrey

**Frank Röder**
Hamburg University of Technology

**Zhanpeng He**
Columbia University

**Tianhe Yu**
Google DeepMind

**Ryan Julian**
Google DeepMind

**K.R. Zentner**
University of Southern California

**Jordan Terry**
Farama Foundation

**Isaac Woungang**
Toronto Metropolitan University

**Nariman Farsad**
Toronto Metropolitan University

**Pablo Samuel Castro**
Google DeepMind
Universite de Montreal
Mila

## Abstract

Meta-World is widely used for evaluating multi-task and meta-reinforcement learning agents, which are challenged to master diverse skills simultaneously. Since its introduction however, there have been numerous undocumented changes which inhibit a fair comparison of algorithms. This work strives to disambiguate these results from the literature, while also leveraging the past versions of Meta-World to provide insights into multi-task and meta-reinforcement learning benchmark design. Through this process we release a new open-source version of Meta-World[1] that has full reproducibility of past results, is more technically ergonomic, and gives users more control over the tasks that are included in a task set.

## 1   Introduction

Reinforcement learning (RL) has made significant progress in real world applications such as the magnetic control of plasma in nuclear fusion [Degrave et al., 2022], controlling the location of stratospheric balloons [Bellemare et al., 2020], or managing a power grid [Yoon et al., 2021]. However, each of these RL agents are limited in their abilities as they are only trained to accomplish a single task. While existing benchmarks have pushed progress in specific areas, such as high-fidelity robotic manipulation, or multi-task and meta learning [Wang et al., 2021, Kannan et al., 2021, Bellemare et al., 2013, Nikulin et al., 2023], they tend to be limited to one specific mode of training and evaluation. Evaluating the ability of reinforcement learning agents to both master diverse skills and generalize to entirely new challenges remains a critical bottleneck.

---

[1]https://github.com/Farama-Foundation/Metaworld/

39th Conference on Neural Information Processing Systems (NeurIPS 2025) Track on Datasets and Benchmarks.

To train RL agents that can accomplish multiple tasks simultaneously, researchers have generally taken one of two approaches. The first is to simultaneously train on multiple tasks with the goal of maximizing accumulated rewards on the same set of tasks. The second approach is to perform meta-learning on a sub-set of tasks where the goal of the RL agent is to learn skills that generalize to a broader set of tasks. Given the significant amount of overlap between these two approaches, Meta-World [Yu et al., 2020b] was proposed to enable effective research on both. Indeed, it has been widely used in foundational multi-task [Yang et al., 2020, Hendawy et al., 2024, Sun et al., 2022, McLean et al., 2025] and meta-RL [Finn et al., 2017, Duan et al., 2017, Grigsby et al., 2024] research, as well as single task RL research [Eysenbach et al., 2022, Nauman and Cygan, 2025].

Since its introduction, however, there have been inconsistencies with the versioning of the benchmark, obfuscating effective comparisons between various algorithms, which we empirically demonstrate below. To address these inconsistencies, we re-engineer the benchmark to facilitate research, benchmarking, customization, and reproducibility. Specifically, our contributions are as follows:

1. We perform an empirical comparison of various multi-task and meta-RL algorithms across past reward functions, highlighting the inconsistencies in cross-version comparisons. We then propose insights for future benchmark design based on the empirical results.

2. We add two new task sets, in addition to the existing task sets, and introduce the ability for users to create custom multi-task or meta RL task-sets.

3. We upgrade the library to be compatible with the latest Gymnasium API [Towers et al., 2024] & Mujoco Python bindings [Todorov et al., 2012], removing the dependencies on the unsupported packages of OpenAI gym [Brockman et al., 2016] & Mujoco-Py.

## 2  Related Works

Reinforcement learning (RL) research has benefited from a variety of standardized benchmarks that enable systematic evaluation and comparison of algorithms. RLBench [James et al., 2020] provides a diverse suite of manipulation tasks in a simulated environment, focusing on robot arm manipulation with varying complexity. Alchemy [Wang et al., 2021] offers a benchmark for testing causal reasoning and exploration in RL settings through chemistry-inspired tasks with compositional structure. More recently, MANISKILL3 [Tao et al., 2025] has expanded the landscape of manipulation benchmarks with a comprehensive set of dexterous manipulation tasks. Kannan et al. [2021] introduces RoboDesk, which is a set of desktop manipulation tasks specifically designed to test generalization capabilities of RL algorithms across different object configurations. However, many of these benchmarks aren't designed to effectively evaluate the performance of multi-task and meta-RL algorithms for robotics tasks due to differing state and action spaces across tasks.

The prominent multi-task RL algorithms evaluated in this work generally followed one of two paths. The first path operates under the assumption that a limitation in multi-task RL is due to difficulties when optimizing multiple tasks simultaneously. This difficulty arises when the gradients of two tasks conflict, defined as when the gradient update from one task overwrites the gradient update from another task. To overcome this limitation, specialty multi-task optimizers, such as PCGrad [Yu et al., 2020a], were proposed. PCGrad takes the gradients from several tasks and if the gradients for two tasks conflict, based on the cosine difference between vectors, then one gradient is projected onto the other.

The second method of training RL agents across multiple tasks is the idea of creating multi-task RL specific architectures. Building upon Soft Actor-Critic (SAC) [Haarnoja et al., 2018], Yu et al. [2020b] proposed the multi-task, multi-head, SAC method (MTMHSAC) that extends SAC by using multiple action heads & an entropy term per task. Following the MTMHSAC method, the Soft-Modularization (SM) architecture was proposed [Yang et al., 2020]. The SM architecture attempts to solve the problem of sharing information between tasks, as it is unclear which information should or should not be shared between tasks. Next, the Parameter Compositional (PaCo) method was developed by Sun et al. [2022]. This architecture posits that there should be a set of parameters shared between tasks, in addition to each task leveraging their own set of parameters. Finally, the Mixture of Orthogonal Experts (MOORE) method was proposed by Hendawy et al. [2024]. This method maintains a shared representation space using a Graham-Schmidt process, which then is used in combination with task specific information to produce actions. One of the limitations of the multi-task approach to training RL agents, is that multi-task RL is only concerned with performance on the current set of tasks. In

order to push the limitations of what RL agents can do, we must design agents that can generalize to unseen tasks.

In order to test the generalization ability of RL agents, the field of meta RL was introduced. In this problem area the goal of the meta RL agent is to be able to learn how to quickly adapt itself to any task within a given task distribution. In general, there are two well known meta-RL algorithms: Model-Agnostic Meta-Learning (MAML) [Finn et al., 2017], and Fast Reinforcement Learning via Slow Reinforcement Learning ($RL^2$) [Duan et al., 2017]. MAML leverages a two stage process for quick adaptation. In the first stage, a set of initial parameters are optimized with a policy gradient to accomplish the current task. In the second stage, the initial parameters are optimized with respect to the final per-task parameters from the insight that the training process of stage one is differentiable with respect to the set of initial parameters. $RL^2$ utilizes a different approach for meta-learning where the policy is conditioned on the entire history of environment interactions, including states, actions, and rewards. This leads to an adaptable policy that learns to adapt with more and more environment interactions. We refer the interested reader to Beck et al. [2025] for a more thorough discussion of the MAML and $RL^2$ algorithms.

# 3 RL Problem Statements

Reinforcement learning (RL) is a popular approach for sequential decision making problems. Each decision occurs at a discrete timestep $t \in 0, 1, 2, .., T$ where $T$ is the maximum number of steps of decision making to occur, called the horizon. At each timestep the agent chooses an action that is applied to an environment. The environment is described as a Markov Decision Process (MDP) $\mathcal{M} = \langle \mathcal{S}, \mathcal{A}, \mathcal{P}, \mathcal{P}_0, \mathcal{R}, \gamma \rangle$, where $\mathcal{S}$ is the set of states, $\mathcal{A}$ is the set of actions, $\mathcal{P} : \mathcal{S} \times \mathcal{A} \times \mathcal{S} \to [0, 1]$ is the probability of transitioning from state $s_t$ to the next state $s_{t+1}$ by applying action $a_t$, $\mathcal{P}_0 : \mathcal{S} \to \mathbb{R}_+$ is the distribution over initial states, $\mathcal{R} : \mathcal{S} \times \mathcal{A} \to \mathbb{R}$ is the reward function, and $\gamma \in [0, 1]$ is a discount factor. In order to map states $s_t$ to actions $a_t$, we learn a policy $\pi(a_t | s_t)$ which is a distribution over actions. The process of interacting with the environment is restricted to a set of episodes which start from sampled initial states from $\mathcal{P}_0$. These episodes last for $T$ steps where the RL agent receives a state $s_t$, chooses an action according to the policy $a_t \sim \pi(\cdot | s_t)$ and the environment transitions to the next state $s_{t+1}$. The goal of the RL agent is to then maximize the expected discounted returns within an episode: $J(\pi) = \mathbb{E}_{\tau \sim \mathcal{P}(\tau)}[\sum_{t=0}^{T} \gamma^t r_t]$ where $\tau$ is a collection of transitions $\tau = \{s_t, a_t, r_t, s_{t+1}\}_{t=0}^{T}$ from an episode called a trajectory, and $\mathcal{P}(\tau)$ is the probability of generating trajectory $\tau$ under the transition function $\mathcal{P}$. Thus the agent must learn to maximize returns along trajectories induced by following the policy $\pi$. While this formulation is technically sound, it does not account for the ability to accomplish multiple tasks in either the multi-task or meta-RL problem settings.

To extend the single task RL formulation to multi-task and meta-RL settings, we must define what a task is. In these settings, we assume that we have access to a distribution of tasks $i \sim p(\mathcal{B})$ that we can sample from. Each task $i$ is itself an MDP $M_i = \langle \mathcal{S}, \mathcal{A}, \mathcal{P}^i, \mathcal{P}_0^i, \mathcal{R}^i, \gamma \rangle$ where each component of the MDP is as previously defined, except that there can be differences across tasks. However, we assume that the set of states $\mathcal{S}$ and set of actions $\mathcal{A}$ are fixed across all tasks.

The objective of the multi-task RL agent is to learn a task-conditioned policy $\pi(a_t | s_t, z_i)$, where $z_i$ is a task descriptor that encodes the ID of the current task $i$. In this work we leverage a one-hot encoding of the task. The objective of maximizing expected returns is also modified to encourage return maximization across all tasks: $J(\pi) = \mathbb{E}_{i \sim p(\mathcal{B})}[\mathbb{E}_{\tau \sim \mathcal{P}_i(\tau)}[\sum_{t=0}^{T} \gamma^t r_t^i]]$ where $r_t^i$ is the reward received at timestep $t$ from task $i$. Unlike meta-RL, in multi-task RL the agent can observe the goal location and the task descriptor $z_i$ in the current state $s_t$.

The objective of the meta-RL agent is fairly similar except for the distributions of tasks. In meta-RL, we aim to endow RL agents with the ability to quickly gather some experiences and leverage them to adapt quickly to the current task. Thus, we split the distribution of tasks into a training and testing distribution $i_{train} \sim p(\mathcal{B}_{train})$ and $i_{test} \sim p(\mathcal{B}_{test})$. During meta-training, the meta-RL agent has access to the train distribution of tasks $p(\mathcal{B}_{train})$, while the agent is evaluated on the meta-test tasks $p(\mathcal{B}_{test})$ which are not seen during training.

# 4  Meta-World

In this section we provide an overview of the Meta-World benchmark. In Appendix A we provide additional information on the different components of Meta-World. Meta-World contains 50 different robotic manipulation tasks for a single Sawyer Robot Arm[2] to solve. Some tasks involve applying force to an object (i.e. *door-close*, *drawer-close*, and *coffee-push*), while some tasks require grasping and manipulating an object (i.e. *pick-place*, and *assembly*), and some combine these two applications of force (i.e. *drawer-open*, *disassembly*). In Appendix A we visualize the included tasks in Meta-World.

## 4.1  Task Sets

To evaluate algorithms in the multi-task setting the MT10 & MT50 task sets are used. In these sets of 10 and 50 tasks, an algorithm is tasked with learning a distribution of tasks with both parametric and task variations. The parametric variations of each task are the random goal or object locations that can be used, where the task variations are the various types of tasks (i.e. *pick-place*, *reach*, *push*). The agent is evaluated on the same tasks that it is trained on.

To evaluate meta-RL algorithms, the ML10 and ML45 benchmarks are used. These benchmarks provide 10 and 45 training tasks respectively for the meta-learning agent to train on, before evaluating it on 5 held-out tasks. In this setting, a meta-learning algorithm needs to learn how to (a) acquire a generalisable set of skills during training, and then (b) compose them in novel ways to adapt to unseen tasks.

The evaluation metric across both multi-task and meta-RL problems is the same, the mean success rate of an agent across all evaluation tasks.

## 4.2  Reward Functions

The original publication of Meta-World [Yu et al., 2020b] created a set of dense reward functions. These rewards, which we will refer to as V1, were designed by creating the *pick-place* reward function, and then modifying the *pick-place* reward function to solve a new task. For the *pick-place* reward function, the rewards guide the agent to: reach towards an object, grasp the object, and move the object to a goal location. To create subsequent rewards this structure would be modified for each new task. For example, for the *reach* task, the reward function would be modified to only include the component of rewards for reaching towards the goal. The portion of the reward function for moving towards an object and gripping the object would be removed. However, at some point these V1 rewards were overwritten by a new set of dense rewards, which we will refer to as V2. These rewards were written with fuzzy constraints that allowed for rewards to be in the range $(0, 10)$ and episodic returns across tasks are from a more narrow distribution. To highlight the differences in rewards, in Figure 1 we plot the per-timestep rewards for the *pick-place* task across the V1 and V2 rewards. For V1, the rewards start slightly negative and then reach a maximum reward for success of around 1200. For V2, the rewards start at zero and reach 10 when the task is solved. We include the full reasoning behind designing these reward functions in Appendix E.

# 5  Empirical Results

Due to historical versioning issues in Meta-World, various works have reported results across different reward functions. These results are not comparable, as the two sets of reward functions have different design philosophies and per-task reward scales as illustrated in Figure 1. In Section 5.1, we provide a comparative analysis on a number of multi-task RL methods over the V1 and V2 reward functions. Similarly, in Section 5.2 we provide meta-RL results using the V1 and V2 rewards. Section 5.3 provides some analysis of our additional task sets MT25 & ML25.

In order to compare the statistical results of each set of experiments we follow recommendations from Agarwal et al. [2021], where we report results over 10 random seeds and use interquartile mean (IQM) to compare results instead of simple averaging. For each graph we plot the IQM performance through training, while also including the 95% confidence interval. In multi-task learning, each method is

---

[2]https://support.rethinkrobotics.com/support/solutions/articles/80000980284-arm-control-system

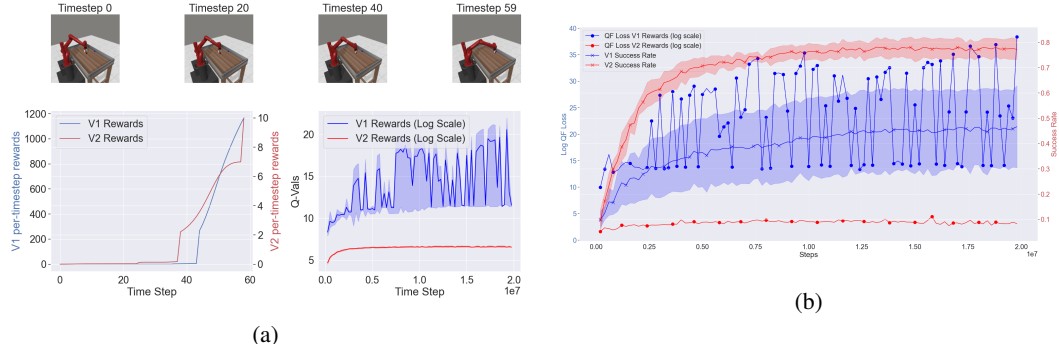

(a)

(b)

Figure 1: (a) Per-timestep rewards for the *pick-place* task from Meta-World. (Left) The Y-axis shows the scale of the V1 rewards, while the right Y-axis shows the scale of the V2 rewards. The right plot shows the Q-Values per-update batch through training on MT10 (log scale). (b) Q-Function loss and mean success rate of the MTMHSAC on MT10. The left y-axis is the Q-Function loss (log scale), while the right y-axis is the mean success rate. Both blue plots are the V1 rewards, while both red plots are the V2 rewards. Plots with circular points are Q-Function loss, while plots with X points are success rates (with 95% CIs).

evaluated for 50 evaluation episodes per task—once for each goal location. In meta-learning, the agent is evaluated after 10 adaptation episodes for 3 episodes for each goal location, per test task. Hyperparameters for each method are gathered from their respective publications. We implement each method into our open-source baselines code base using JAX (Bradbury et al. [2018]) [3].

## 5.1 Multi-task RL Results

In this section we outline the MT10 and MT50 results across both the V1 and V2 reward functions, while also corroborating, and refuting, recent results in multi-task RL. In Table 1 we report the performance of the methods that are commonly incorrectly compared due to aforementioned versioning issues. The *Pub* columns indicate values gathered directly from the publication of the method, while the other columns are performance values that we gather using our own implementations of each method.

In Figure 2a and Figure 2b we report the IQM learning curves for MT10 and MT50, respectively, trained on each of the selected multi-task RL methods: MTMHSAC, SM, MOORE, PaCo, and PCGrad.

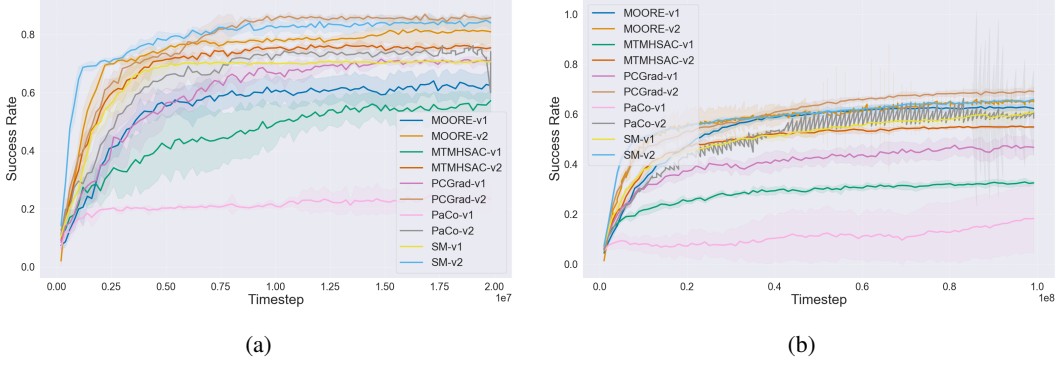

(a)

(b)

Figure 2: Effects of reward functions on selected multi-task algorithms on (a) MT10 and (b) MT50.

When examining Figure 2a, we find that algorithms generally struggle with the V1 rewards on MT10. This seems to indicate that the V1 rewards are more difficult to optimize as the algorithms have lower

---

[3]https://github.com/rainx0r/metaworld-algorithms

| Algorithm | Pub MT10 | MT10 V1 | MT10 V2 | Pub MT50 | MT50 V1 | MT50 V2 |
|---|---|---|---|---|---|---|
| SM | 71.8 | **71.4** | 84.9 | 61.0 | **60.6** | 65.8 |
| PaCo | 85.4 | 26.2 | **73.6** | 57.3 | 18.6 | **58.4** |
| MOORE | 88.7 | 61.4 | **83.2** | 72.9 | 61.2 | **72.0** |

Table 1: Comparing results from a recent multi-task RL publication [Hendawy et al., 2024], to the results we empirically validate across the V1 & V2 reward functions. Pub MT10 and Pub MT50 are from Hendawy et al. [2024], while the V1 and V2 columns are of our own implementations. Bolded results indicate the reward function that was used to produce the result for that specific method.

performance when compared with the V2 rewards. In the MT10 V2 setting, our results indicate that the V2 reward function is somewhat easier to optimize as the performance of each tested algorithm increases compared to the V1 results. Here we find that PCGrad and SM are again the two top performing methods.

Moving on to the MT50 setting, in Figure 2b, we observe much the same: final performance on the V1 rewards is much lower than on the V2 ones, even more so than on MT10, indicating that the optimization difficulties not only continue as the number of tasks grows but even become exacerbated with more tasks. Therefore, perhaps owing to their respective methods of alleviating said issue, SM and PCGrad continue to outperform the rest of the baselines, though much less so on the V2 rewards, where MOORE is on par with SM.

Through our empirical results and analysis, we find that the implementation of the V2 reward functions does follow the design principle that was intended. In Figure 1b we find that when training the MTMHSAC agent on the V1 & V2 rewards, that the Q-Function loss of the V2 trained MTMHSAC agent is much lower than the V1 MTMHSAC, and the V2 rewards lead to a higher success rate overall for the agent. In fact, we find that all tested algorithms perform better on the V2 rewards than the V1 rewards. This seems to show that in multi-task RL the capability of the Q-Function to model state-action values is tied to the overall success rate of the agent, which agrees with recent work [McLean et al., 2025].

In this section we have empirically demonstrated that the V2 rewards enable multi-task RL agents to achieve higher success rates than when trained on the V1 rewards, likely due to the Q-Function's enhanced ability to accurately model state-action functions when rewards have similarly scales across tasks. This echoes the findings of Hessel et al. [2019]: reward functions should maintain similar scales for episodic returns across all tasks to facilitate effective multi-task learning.

These insights open several promising directions for future benchmarks. These new benchmarks could evaluate agents on their ability to (a) follow language instructions where rewards would need to maintain consistent scaling regardless of instruction complexity, (b) perform multi-stage tasks that combine primitive actions (e.g., picking up object 1 then pushing object 2) with well-calibrated composite rewards, or (c) learn effectively across heterogeneous reward types (human feedback, sparse and dense rewards) while controlling for scale variations. Our standardized codebase and explicit versioning will enable researchers to build these extensions while avoiding the inconsistencies we've identified.

## 5.2 Meta-RL Results

Next we examine the results of meta-RL algorithm performances across the V1 and V2 reward functions available in Meta-World. In Figure 3a and 3b we report the IQM across the ML10 and ML45 task sets, plotting the test success rate through training. We implement MAML [Finn et al., 2017] and $RL^2$ [Duan et al., 2017] as they are the two representative algorithms that many meta-RL approaches build on.

From Figure 3a & 3b we find that there is statistically no difference between MAML-V1, MAML-v2, and $RL^2$-V2. We posit that this is likely due to the meta-RL algorithms learning from experience in a different way than the multi-task algorithms. The multi-task algorithms we explored were based on SAC, which is predominantly a value-based algorithm where Q-learning ability correlates with final performance in this setting [McLean et al., 2025]. Meanwhile, in our MAML and $RL^2$ implementations we use policy gradient methods using returns computed with Generalised Advantage Estimation [Schulman et al., 2015], namely TRPO [Schulman et al., 2015] and PPO [Schulman

et al., 2017] respectively, and more importantly, use the standard linear feature baseline proposed by Duan et al. [2016]. Since this baseline does not use gradient descent and instead directly fits a linear function onto the collected trajectories at each optimization step, the advantages used in policy optimization are always of a low magnitude and reasonable variance. However, the performance of $RL^2$ on the V1 rewards shows a large performance drop and matches the performance of $RL^2$ in Yu et al. [2020b]. This is likely due to the raw rewards being used in the observation as they are not normalized, leading to optimization challenges.

The modest overall performance of these algorithms, however, suggests that there is still a lot of room for meta-RL research that can learn efficiently under the constraints of Meta-World. The disjoint task space and compositional complexity still prove to be a challenge for algorithms aiming to effectively generalize Walker et al. [2023]. Recent work that demonstrates gains in other meta-rl settings using classification losses and transformer models echoes this sentiment, focusing on the more traditional setting of ML1 rather than tackling ML10 and ML45 Grigsby et al. [2024], highlighting the need for new ideas in this area.

This, however, implies a gap between the parametric variation setting of ML1, where current meta-rl methods struggle less, and the challenging non-parametric task distribution with relatively few task classes of ML10 and ML45. Future meta-RL benchmarks should therefore aim to interpolate between the two, providing a wider task distribution with greater compositional diversity. One such direction could be incorporating different robot morphologies and explicitly exploring transfer across varying embodiments, along the lines of Bohlinger et al. [2024] & McLean et al. [2024]. Such improvements would better reveal the comparative advantages of different meta-RL approaches while maintaining the consistent reward scaling principles identified in this work.

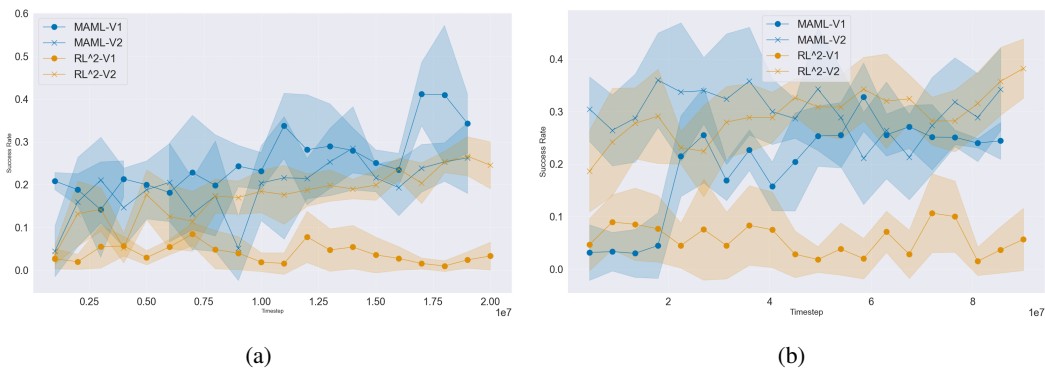

Figure 3: IQM results over (a) ML10 and (b) ML45. Plotted results are the testing success rates.

## 5.3   Results over various task set sizes

Finally, in Figure 4, we explore the effects of introducing an additional task set to the Meta-World benchmark. We discuss the construction of MT25 & ML25 in Appendix B. The introduction of MT25/ML25 and customizable task sets offers significant practical advantages for researchers. Conducting experiments on MT50/ML45 can be computationally expensive, where MT25/ML25 provides a middle ground that reduces computational costs by approximately 50% compared to MT50/ML45 while still offering more robust insights than the smaller MT10/ML10 benchmark. Our own experiments had walltimes of: MT10 $\sim$ 6 hours, MT25 $\sim$ 12 hours, and MT50 $\sim$ 25 hours on an AMD Epyc 7402 24-Core Processor with an NVIDIA A100 PCI GPU. This efficiency allows researchers to conduct preliminary experiments and algorithm comparisons more rapidly, reserving full MT50/ML45 evaluations for finalized methods. Perhaps most importantly, the ability to create custom task sets of arbitrary size enables researchers to design controlled experiments that isolate specific learning challenges. For instance, researchers can construct task sets with carefully selected similarities to study knowledge transfer, or create sets with balanced difficulty distributions to fairly evaluate different algorithmic approaches. This customization capability significantly expands the types of research questions that can be systematically addressed using the Meta-World platform.

In Figure 4a, we find that when doing multi-task learning across benchmarks of different sizes, that the performance of the MTMHSAC agent drops. McLean et al. [2025] recently found that increasing

the number of tasks for a given parameter count aids in mitigating plasticity loss, so this performance drop is likely a network capacity issue. In Figure 4b, we find that performance is approximately identical regardless of the size of the task set.

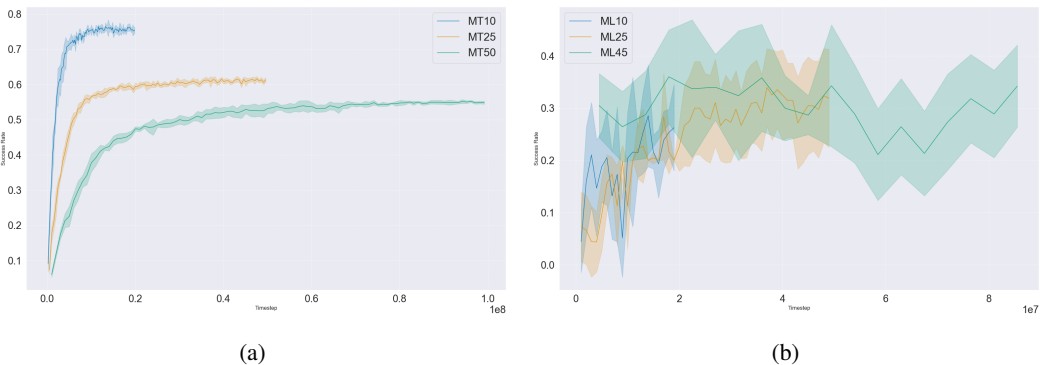

(a)                                    (b)

Figure 4: (a) Effects of various task set sizes on the performance of the MTMHSAC agent. (b) Effects of various task set sizes on the performance of the MAML agent.

# 6 Overall Benchmark Improvements

**Reasoning for New Version of Meta-World**   To address inconsistencies between reward versions, our new implementation takes a preservation-based approach rather than introducing yet another reward system. Instead of creating a V3 reward function, we explicitly maintain both V1 and V2 reward functions as selectable options through a standardized API.

**Gymnasium Integration**   By aligning Meta-World with the Gymnasium standard rather than maintaining its custom environment implementation, we enable researchers to leverage the full ecosystem of Gymnasium tools, infrastructure, and environment creation as highlighted in Figure 5. Additional considerations are available in Appendix A.

# 7 Conclusion

In this work we highlighted a discrepancy in the multi-task and meta-RL literature due to issues with the software versioning of the Meta-World benchmark. Through empirical evaluation, we demonstrated that inconsistent reward functions led to incorrect conclusions about algorithm performance, with multi-task RL and meta-RL methods showing significant sensitivity to reward scaling differences.

This work highlights the need for explicit versioning of software benchmarks that researchers should report in their works. Had this versioning been in place throughout the history of Meta-World, the discrepancies between results likely would not have happened. In addition to software versioning, we have also highlighted the need for better practices when making large changes to an existing benchmark that may affect results. Lastly, we find that simply reporting results from a previous work, rather than running the method (through one's own implementation, or open sourced code), likely also contributed to

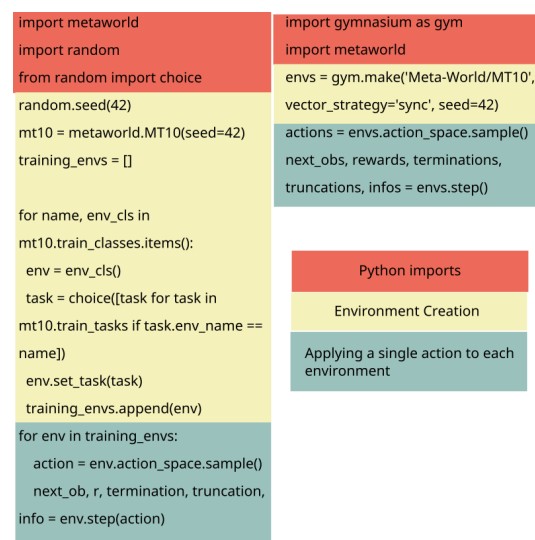

Figure 5: Process for using Meta-World in previous versions (left), and our updated version (right).

these issues with Meta-World. Thus, we believe
that users should be running their own baselines
rather than copying numbers from previous works.

Due to the issues that we have highlighted in this work, we argue that benchmark development should prioritize transparent versioning, reward function consistency, and explicit documentation of design decisions that affect performance. Beyond addressing versioning issues, next-generation multi-task and meta-RL benchmarks should explore greater task diversity, compositional complexity, and cross-embodiment transfer challenges to better differentiate algorithmic advances. By building standardized evaluation protocols with these principles in mind, the community can ensure that reported improvements reflect genuine algorithmic progress rather than artifacts of implementation differences or benchmark design variations.

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

# A   Meta-World

Meta-World is a collection of robotic manipulation tasks with a shared observation & action space designed to learn multi-task or meta-RL policies on a collection of tasks. In previous works, such as Yang et al. [2020], there is some discussion of 'fixed' and 'conditioned' versions of Meta-World. This distinction does not exist in the current version of Meta-World, though it may have in earlier versions of Meta-World.

## A.1   Additional Quality of Life Improvements

### A.1.1   Quality of Life Improvements

Figure 5 shows the new streamlined benchmark creation process which significantly reduces the barrier to entry for new users. Previously, environment instantiation required detailed knowledge of Meta-World's internal architecture and multiple configuration steps. With the new gym.make interface, users can create environments using a single, intuitive command, following the familiar Gymnasium paradigm.

### A.1.2   Full Reproducibility of Past Results

Maintaining backward compatibility while introducing these improvements ensures that researchers can reproduce and build upon previous Meta-World results. We have carefully preserved the core functionality and task definitions from earlier versions while adding new capabilities. This approach allows researchers to verify their implementations against established benchmarks and directly compare their results with previous work.

## A.2   Tasks

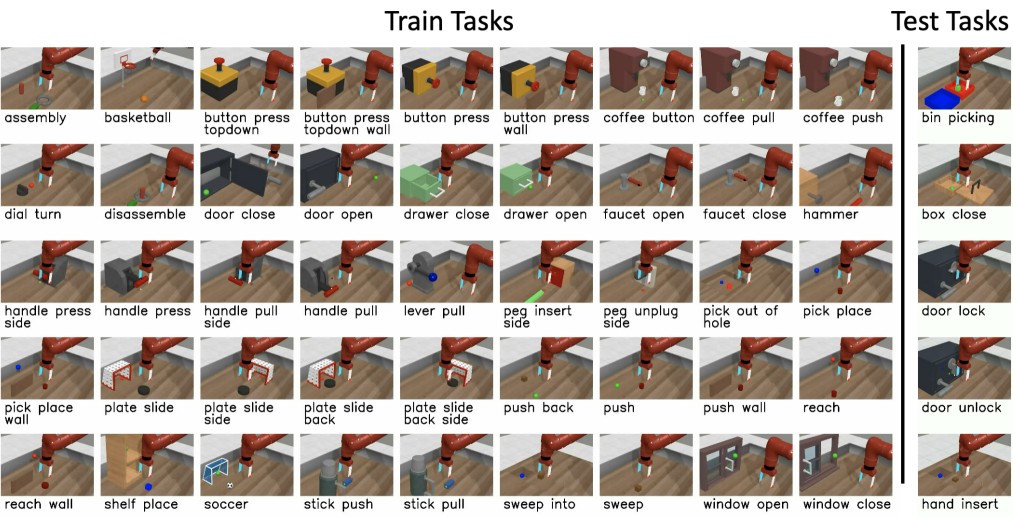

Figure 6: Tasks available within Meta-World. Image from Yu et al. [2021].

## A.3   Reward function

For each of the 50 available tasks in Meta-World, there are 2 dense reward functions. The first reward function, V1, was used to produce the results in (cite CoRL paper). The second reward function, V2, was used to produce the results in (cite Meta-World arxiv paper). We believe that by exposing both the V1 and V2 reward functions to the user, that users can more accurately compare the results of their algorithms to other published results. This would also allow for users to evaluate their algorithms on variations of Meta-World benchmarks that may show how well their methods work in various settings.

### A.4 Observation & Action Spaces

The observation space and action spaces are inherited from the Gymnasium standard for continuous spaces. The action space is a 4-tuple where the first three elements of the 4-tuple are the desired end-effector displacement, and the last element is the position of the gripper fingers.

The observation space is designed to be shared across all 50 available tasks in Meta-World. Thus, they must contain information for all of the different task scenarios. In order to be used across all tasks, each observation in Meta-World is 39-dimensional. Positions (zero indexed, inclusive) in the observation represent the following attributes:

- $[0 : 2]$: the XYZ coordinates of the end-effector
- $[3]$: a scalar value that represents how open/closed the gripper is
- $[4 : 6]$: the XYZ coordinates of the first object
- $[7 : 10]$: the quaternion describing the spatial orientations and rotations of object #1
- $[11 : 13]$: the XYZ coordinates of the second object
- $[14 : 17]$: the quaternion describing the spatial orientations and rotations of object #2

Both of the objects XYZ coordinates and quaternions can exist (i.e. in the *coffee-push* task), or if only one object exists the second object XYZ coordinates and quaternion are zeroed out. List A.4 only outlines an observation of size 18. The observation at time $t$ is stacked with the observation from time $t - 1$, and the XYZ coordinates of the goal, to complete the 39-dimension observation vector. Meta-World is thus a goal-conditioned task. In multi-task reinforcement learning the goal is visible to the agent, while in meta-reinforcement learning the goal is zeroed out.

### A.5 Tasks & Task-Sets

```
import gymnasium as gym
import metaworld
# this registers the Meta-World environments with Gymnasium

# create a MT1 environment object
envs = gym.make('Meta-World/MT1', env_name='reach-v3', seed=42)

# create a MT10 environment object
envs = gym.make('Meta-World/MT10', vector_strategy='sync', seed=42)

# create a MT50 environment object
envs = gym.make('Meta-World/MT50', vector_strategy='sync', seed=42)

# create a custom benchmark environment object
envs = gym.make('Meta-World/MT-custom', env_names=['env1-v3', 'env2-v3
                                    ', ..., 'envN-v3'], vector_strategy
                                    ='sync', seed=42)
```

Figure 7: Multi-task environment creation examples. Note that multi-task algorithms are evaluated on the training environments.

## B  MT25 and ML25 Construction

To create a new task set, we used the results from training on MT50. To create the training task set of 25 tasks, we select 12 tasks that are solved in MT50 and 13 tasks that are not solved.

## C  Replication

In Table 2 we report the results of our own implementations of PCGrad, SM, PaCo, and MOORE compared to the results found in their respective publications. We find that SM, PaCo, and MOORE

```python
import gymnasium as gym
import metaworld
# this registers the Meta-World environments with Gymnasium

# create ML1 environment objects
train_envs = gym.make('Meta-World/ML-1-train', env_name='reach-v3',
                                    seed=42)
test_envs = gym.make('Meta-World/ML-1-test', env_name='reach-v3', seed
                                    =42)

# create ML10 environment objects
train_envs = gym.make('Meta-World/ML10-train', vector_strategy='sync',
                                    seed=42)
test_envs = gym.make('Meta-World/ML10-test', vector_strategy='sync',
                                    seed=42)

# create a ML45 environment object
train_envs = gym.make('Meta-World/ML45-train', vector_strategy='sync',
                                    seed=42)
test_envs = gym.make('Meta-World/ML45-test', vector_strategy='sync',
                                    seed=42)

# create a custom ML benchmark environment object
train_envs = gym.make('Meta-World/ML-custom', env_names=['env1-v3', '
                                    env2-v3', ..., 'envN-v3'],
                                    vector_strategy='sync', seed=42)
test_envs = gym.make('Meta-World/ML-custom', env_names=['env1-v3', '
                                    env2-v3', ..., 'envN-v3'],
                                    vector_strategy='sync', seed=42)
```

Figure 8: Meta-learning environment creation examples. Note that meta-learning tasks have a distinction between training and testing environments.

| Architecture | MT10 (Ours) | MT10 (Pub) | MT50 (Ours) | MT50 (Pub) |
|---|---|---|---|---|
| PCGrad [Yu et al., 2020a] | 70.5 | 90 | 45.8 | 70 |
| Soft Modularization [Yang et al., 2020] | 71.4 | 71.8 | 60.6 | 61.0 |
| PaCo [Sun et al., 2022] | 73.6 | 71.6 | 58.9 | 57.3 |
| MOORE [Hendawy et al., 2024] | 83.2 | 88.7 | 72.0 | 72.9 |

Table 2: Replication of selected MTRL results. For the PCGrad & SM results, these results use the V1 rewards. Both PaCo and MOORE leverage the V2 rewards. We note that our PCGrad results are not replicating the original PCGrad results, however it is inline with the values used in previous work [Sodhani et al., 2021].

are all approximately matching the performance of the published results. We find that we cannot replicate the published results from PCGrad, however the results we report are similar to the results reported by Sodhani et al. [2021]. This is likely due to the authors using a version of Meta-World that is no longer available since the PCGrad work was published around the time that Meta-World was originally developed.

# D   Raw Results

# E   Reward Function Designs

## E.1   Meta-World v1 Reward Functions

The original published version of Meta-World had a single primary reward function, written for the pick-place-wall task, which was then modified to perform other task. This reward function

```python
import gymnasium as gym
import metaworld
# this registers the Meta-World environments with Gymnasium
from metaworld.evaluation import evaluation, metalearning_evaluation,
                                    Agent, MetaLearningAgent

# Multi-task
# Explicitly subclassing is optional but recommended
class MyAgent(Agent):
    def eval_action(self, observations):
        ... # agent action selection logic

agent = MyAgent()
envs = gym.make('Meta-World/MT10', vector_strategy='sync', seed=42)
mean_success_rate, mean_returns, success_rate_per_task = evaluation(
                                    agent, envs)

# Meta-Learning
# Explicitly subclassing is optional but recommended
class MyMetaLearningAgent(MetaLearningAgent):

    def eval_action(self, observations):
        # agent action selection logic
        ...

    def adapt_action(self, observations):
        # agent action selection logic
        ...
        # this can also return values needed for adaptation like
                                    logprobs

    def adapt(self, rollouts) -> None:
        # Adaptation logic here, this should mutate the agent object
                                    itself
        ...

agent = MyMetaLearningAgent()
test_envs = gym.make('Meta-World/ML10-test', vector_strategy='sync',
                                    seed=42)
mean_success_rate, mean_returns, success_rate_per_task =
                                    metalearning_evaluation(agent,
                                    test_envs)
```

Figure 9: Sample code for evaluating agents on Meta-World.

guided the policy through performing a specific sequence of steps using a combination of geometric primitives. Specifically, it contained the following sequence:

1. Move the gripper above the target object and open the gripper.

2. Once the gripper was directly above the target object, move the gripper down.

3. Once the gripper was within a particular distance of the target object, close the gripper. Remember this step has occurred, to keep the gripper closed in later steps.

4. Move the gripper to height that will avoid any walls.

5. Move the gripper above the goal location.

6. Lower the gripper to the goal location.

This reward function was tuned for different tasks by removing unnecessary steps. The resulting reward functions worked reasonably well for some tasks, but for some tasks optimizing the reward function would never result in a policy with a high success rate.

Table 3: V1 MT10 Results

| Algorithm | Result |
|---|---|
| MTMHSAC | $46.88 \pm 22.69$ |
| Soft Modules | $71.41 \pm 2.08$ |
| MOORE | $61.44 \pm 7.10$ |
| PaCo | $26.18 \pm 7.54$ |
| PCGrad | $70.49 \pm 2.16$ |

Table 4: V2 MT10 Results

| Algorithm | Result |
|---|---|
| MTMHSAC | $77.26 \pm 5.64$ |
| Soft Modules | $84.88 \pm 4.51$ |
| MOORE | $83.42 \pm 7.93$ |
| PaCo | $72.26 \pm 7.44$ |
| PCGrad | $85.52 \pm 1.72$ |

### E.2 Meta-World v2 Reward Functions

As part of the v2 update for Meta-World, a new set of reward functions were written. This would involve 50 (mostly) unique reward functions, with the following objectives:

1. For each task, it should be possible for PPO to produce a policy with a high success rate using 20 million timesteps using the reward function.

2. The reward functions should be minimally opinionated. In particular, there should be no motions (such as moving up to avoid obstacles), that are obviously a result of the reward function and not the success criterion.

3. The reward functions should span a minimal difference in scale. Early work in Multi-Task RL indicated that differences in reward scales between tasks was a major difficulty, but we wanted to de-emphasize that aspect of the challenge.

4. The reward functions should be Markovian (they should not depend on prior states or hidden variables).

We found it difficult to simultaneously achieve the first three objectives perfectly. Ultimately, the reward functions we wrote were able to perform fairly well on all three points:

1. PPO is able to reach roughly 90% success rate after 20 million timesteps in at least 45/50 tasks (90% of the tasks).

2. The reward functions contain significant information about "bottleneck" states (such as if an object is held in the gripper), but contain no explicit temporal information, and minimal information about e.g. avoiding obstacles.

3. The reward functions all scale approximately two orders of magnitude, from 0.1 to 10. We found that this performed significantly better than having a maximal reward each timestep of 1.

### E.3 Fuzzy-Logic Reward Function Paradigm

In other to achieve the above objectives, we experimented with a new paradigm for writing reward functions using fuzzy logic. In this paradigm, each reward function consists of a set of "fuzzy" geometric constraints, which are then combined using "fuzzy conjunction" (using the Hamacher product), or "fuzzy disjunction" by taking a weighted sum. The exact method of computing fuzzy geometric constraints varies from constraint to constraint, but the most frequent is to use a custom sigmoid function of the difference between two coordinates (e.g. $\sigma(x_1 - x_2)$). This sigmoid function has a long tail in the negative region, (when the constraint is not satisfied), and quickly saturates to 1 in the positive region (when the constraint is satisfied). In every case, the fuzzy logic value representing a constraint lies in the range $(0, 1]$, where 1 corresponds to the constraint being satisfied.

Table 5: V1 MT50 Results

| Algorithm | Result |
|---|---|
| MTMHSAC | $31.92 \pm 2.67$ |
| Soft Modules | $60.59 \pm 3.86$ |
| MOORE | $61.772 \pm 2.66$ |
| PaCo | $18.58 \pm 15.38$ |
| PCGrad | $45.84 \pm 5.93$ |

Table 6: V2 MT50 Results

| Algorithm | Result |
|---|---|
| MTMHSAC | $54.90 \pm 1.07$ |
| Soft Modules | $65.78 \pm 1.94$ |
| MOORE | $71.99 \pm 2.93$ |
| PaCo | $58.88 \pm 4.64$ |
| PCGrad | $69.49 \pm 2.67$ |

After computing the fuzzy logic reward function, we rescaled them by a factor of 10, which appears to have improved the performance of PPO.

We found that this paradigm had the following advantages:

1. We could improve training performance on a particular task by adding additional constraints to its reward function until PPO was able to achieve our desired training performance.

2. Because our fuzzy constraint functions output 1 when satisfied, satisfied constraints don't affect the reward function in the region where they are satisfied. This allows the reward functions to be less opinionated using e.g. keep-out regions that don't bias the reward function outside of a small radius around the keep-out region.

3. We could maintain a relatively consistent structure and scale between each of the 50 tasks by using similar equivalent constraints, and because the fuzzy logic naturally scaled from 0 to 1.

Table 7: V1 ML10 Results

| Algorithm | Result |
|---|---|
| MAML V1 | $39.58 \pm 16.11$ |
| RL2 V1 | $6.53 \pm 7.74$ |
| Amago-2 V1 | $3 \pm 4$ |

Table 8: V2 ML10 Results

| Algorithm | Result |
|---|---|
| MAML V2 | $30.1 \pm 11.45$ |
| MAML V2 (no baseline) | $2.00 \pm 3.46$ |
| RL2 V2 | $25.7 \pm 8.92$ |
| Amago-2 V2 | $6 \pm 4$ |

Table 9: V1 ML45 Results

| Algorithm | Result |
|---|---|
| MAML V1 | $26.67 \pm 7.67$ |
| RL2 V1 | $9.71 \pm 12.31$ |
| Amago-2 V1 | $8 \pm 4$ |

Table 10: V2 ML45 Results

| Algorithm | Result |
|---|---|
| MAML V2 | $32.4 \pm 14.47$ |
| RL2 V2 | $36.76 \pm 13.90$ |
| Amago-2 V2 | $11 \pm 3$ |

