# OpenReview forum: "Meta-World+: An Improved, Standardized, RL Benchmark"
_NeurIPS.cc/2025/Datasets_and_Benchmarks_Track — NeurIPS 2025 Datasets and Benchmarks Track poster_

### Official Review · Reviewer_xUrJ · 2025-06-02

**Rating:** 5
**Confidence:** 4

**Summary:**

This paper revisits the widely used Meta-World benchmark for evaluating multi-task and meta-reinforcement learning (meta-RL) algorithms, highlighting inconsistencies caused by undocumented changes and reward function versioning. To address these issues, the authors present Meta-World+, a re-engineered version of the benchmark that emphasizes reproducibility, customization, and compatibility with modern RL libraries (e.g., Gymnasium and MuJoCo).

**Additional Feedback:**

Appendix D explains the motivation and structure of V1 and V2 reward functions, but the actual code-level implementation is not described in detail. Suggestion: Provide a concrete example or pseudocode for how V1 and V2 reward functions differ for a specific task (e.g., pick-place), along with guidance on how users can modify or extend them.

**Dataset Code Accessibility:**

Partly

**Ethical Considerations:**

No, there are no or only very minor ethics concerns

**Final Justification:**

Thanks for authors comments and my comments are carefully addressed. The statistical report gives more rational. I have raised my score.

**Limitations Weaknesses:**

The paper is not proposing a new RL algorithm, architecture, or training procedure.

Its contributions are entirely benchmark-focused, aimed at improving experimental reproducibility and task design clarity.
This places it closer to a systems or benchmark paper, which is valuable but must be judged on practical utility and methodological rigor rather than algorithmic novelty. While it exposes meaningful flaws in previous uses of Meta-World, many components (e.g., reward reimplementation, compatibility with Gymnasium) are logical updates rather than fundamentally novel insights.

The idea of standardizing reward scales and exposing task composition is important, but not unprecedented.

Here are some detailed comments:

1.  Section 5.2, Figure 3 (ML10/ML45 results)
Weakness: While the paper highlights inconsistencies in reward scaling for meta-RL tasks, it offers limited analysis or innovation in addressing the fundamental challenges of meta-RL generalization. The experiments focus only on MAML and RL², two older baselines, without testing more recent transformer-based or hybrid methods that claim better compositional generalization (e.g., AMAGO-2 [Grigsby et al., 2024] is cited but not tested).
Also, meta-learning can be metric-based , model-based, optimization-based, which approach does meta-RL use? Some more detailed explanation of method itself need to describe. If it is optimization-based, how does the bi-level optimization algorithm minimizes?

Suggestion: Extend the meta-RL evaluation to include state-of-the-art architectures, especially those that explicitly address task compositionality or memory bottlenecks. Moreover, incorporate fine-grained ablation studies to explore which features of the V2 reward function contribute most to the observed performance improvements.


2.  Section 4.1, 5.3; Appendix B (MT25/ML25 construction)
Weakness: Although MT25/ML25 are introduced to offer intermediate difficulty levels, the construction methodology (12 “solved” + 13 “unsolved” tasks) from MT50 (Appendix B) lacks formal criteria or justification. It’s unclear whether these new sets offer meaningful compositional or representational diversity beyond what MT10/ML10 provide.

Suggestion: Provide quantitative metrics for task diversity, such as task-to-task similarity (e.g., based on reward function features, state transitions, or skill primitives). A clustering or coverage analysis would help justify the value of MT25/ML25 beyond computational convenience.


3. Section 5.2 (Future Directions), Section 6 (Conclusion)
Weakness: Although the paper briefly mentions future directions involving cross-embodiment transfer (e.g., different robot morphologies), the current version of Meta-World+ is still restricted to the Sawyer Arm, limiting the benchmark’s relevance to generalizable robotics or sim-to-real studies.

Suggestion: Integrate multi-embodiment support or at least provide a roadmap and simulation stubs for extending Meta-World+ to include alternative robot platforms (e.g., Franka Panda, UR5). This would significantly increase its utility for embodied RL and domain adaptation research.


4. Section 5.2

Weakness: The claim that reward scale differences have less impact on policy-gradient-based meta-RL methods (vs. value-based MTRL) is plausible but insufficiently substantiated. The paper notes that the linear feature baseline used by TRPO/PPO might mitigate this (Lines 231–235), but no controlled ablations are presented.

Suggestion: Conduct a controlled experiment where normalization or baseline choices are varied within a single meta-RL method (e.g., PPO with/without advantage normalization). This would clarify whether the reward scale is truly less influential for meta-RL, or whether the observed robustness is a side effect of algorithm design choices.

5. Checklist Item #10 (Broader Impacts)

Weakness: The authors claim the work has “no direct societal impact,” which overlooks the central role of benchmarks in shaping research priorities and evaluations in reinforcement learning. Benchmark biases (e.g., overfitting to synthetic task sets) can inadvertently stall progress toward real-world RL deployment.

Suggestion: Add a brief section in the conclusion or discussion addressing how benchmark design (including reward design and task composition) might influence the development of generalizable, safe, and interpretable RL agents. This helps contextualize the broader significance of the work.

6. Section 5.1, Table 1; Section C (Replication)

Weakness: While the paper reimplements prior algorithms for consistency, there is discrepancy in replication (e.g., PCGrad results are not reproducible, Table 2). The authors attribute this to an unavailable Meta-World version, but no detailed forensics or intermediate diagnostics are provided to support this claim.

Suggestion: Include controlled sanity checks, such as confirming gradient alignment behavior or reward traces for key tasks, to demonstrate equivalence between reimplementations and reported behaviors. Alternatively, offer version-locked containers or scripts to reproduce the published results faithfully and verify discrepancies.

**Strengths Contributions:**

Meta-World has become a widely used benchmark in multi-task and meta-RL research.

This paper addresses a critical and widely under-discussed issue: inconsistent reward versioning, which has led to invalid comparisons across papers over several years.

By offering reproducible results, API modernization (Gymnasium), and custom task set generation, the authors provide meaningful infrastructure upgrades likely to be adopted by the RL community.

Added two new task sets: MT25 (multi-task) and ML25 (meta-RL), offering a middle ground in computational cost and benchmark complexity.

The emphasis on IQM metrics, 10-seed evaluations, and open-sourcing of training scripts/code suggests good reproducibility standards.

Users can now define custom task sets, enabling controlled experimental design and better flexibility.

---

> ### Author Rebuttal · Authors · 2025-07-29
>
> We thank Reviewer xUrJ for their thorough review and comments to improve our work. We would like to highlight that our goal with Meta-World+ is to enable scientific rigor that enables cumulative research progress and reduces validation barriers.
>
> > While the paper highlights inconsistencies in reward scaling for meta-RL tasks, it offers limited analysis or innovation in addressing the fundamental challenges of meta-RL generalization
>
> The Meta-RL algorithms that we have chosen are both gradient based algorithms. The reply to reviewer KifD includes data from Amago-2.
>
> > Section 4.1, 5.3; Appendix B (MT25/ML25 construction) Weakness: Although MT25/ML25 are introduced to offer intermediate difficulty levels, the construction methodology
>
> We apologize for the prominent claim of introducing new task sets of MT25 & ML25, our intention was to show that there is a meaningful middle ground between 10 tasks and 50/45 tasks, and that we have exposed the ability for users to create custom task sets to explore the effects of different tasks used during training or testing.
>
> We ran some rudimentary experiments for task complexity based on episodic returns of an expert policy. For this experiment we capture the episodic returns, with an episode length of 500, for each timestep the policy is interacting with the task. Once the task is solved, we extend the success reward for the remaining timesteps to ensure that each episode is the same length to ensure fair comparisons between episodic returns. Using this method we then calculate the mean episodic returns of all tasks in the task set (MT10, MT25, or MT50).  Using this mean episodic return, we then split the tasks into tasks that produced returns above the mean (i.e. the task was solved early in the episode and is ‘easy’), and tasks that produced returns below the mean (i.e. the task required more steps to be solved and is ‘hard’). Using this method we produce the following data:
>
> MT10: 6 ‘easy’ tasks, 4 ‘hard’ tasks -> 60% easy, 40% hard
>
> MT25: 15 ‘easy’ tasks, 10 ‘hard’ tasks -> 60% easy, 40% hard
>
> MT50: 31 ‘easy’ tasks, 19 ‘hard’ tasks -> 62% easy, 38% hard
>
> From this data, we find that we have maintained the approximate level of difficulty of MT10 and MT50 in our new task set MT25.
>
> > Section 5.2 (Future Directions), Section 6 (Conclusion) Weakness: Although the paper briefly mentions future directions involving cross-embodiment transfer
>
> We appreciate the reviewer’s desire to support multi-embodiment as we also feel that it is an important avenue for future research. In the next comments, we have included an example of how to integrate the Franka Emika Panda arm from Mujoco Menagerie into Meta-World+.
>
> > Weakness: The claim that reward scale differences have less impact on policy-gradient-based meta-RL methods (vs. value-based MTRL) is plausible but insufficiently substantiated.
>
> We apologize for the confusion in these remarks in the text. We note that the performance of MAML is less sensitive to rewards due to the use of a feature baseline, however this is not the case with RL2 as the raw rewards are passed to the RNN module for optimization. We will update the manuscript to provide more clarity on this. We have also produced some results ablating the linear feature baseline as the reviewer suggested. The ablation of the baseline changes the performance profile of MAML completely, which confirms our comment that the linear feature baseline makes MAML performance agnostic to the choice of rewards.
>
> ### ML10 V1 Results
> | Algorithm | Result | Std Dev |
> | ------------- |:-------------:|:-------------:|
> | MAML | 39.58 | 16.11 |
> | MAML (no baseline) | 4.00 | 2.83 |
>
> ### ML10 V2 Results
> | Algorithm | Result | Std Dev |
> | ------------- |:-------------:|:-------------:|
> | MAML | 30.1 | 11.45 |
> | MAML (no baseline) | 2.00 | 3.46 |
>
> ### ML45 V1 Results
> | Algorithm | Result | Std Dev |
> | ------------- |:-------------:|:-------------:|
> | MAML | 26.67 | 7.67 |
> | MAML (no baseline) | 0 | 0 |
>
> ### ML45 V2 Results
> | Algorithm | Result | Std Dev |
> | ------------- |:-------------:|:-------------:|
> | MAML | 32.4 | 14.47 |
> | MAML (no baseline) | 0 | 0 |
>
> > Checklist Item #10 (Broader Impacts) Weakness: The authors claim the work has “no direct societal impact,” which overlooks the central role of benchmarks
>
> We thank the reviewer for their insightful comments about benchmark design influencing the development of generalizable, safe, and interpretable RL agents. We will add some commentary of these points in the camera ready version of the paper along the lines of:
>
> While creating and maintaining a benchmark is an important task, we also note that benchmarks can cause researchers to focus on the improvement of benchmark scores rather than focusing on real-world deployments of reinforcement learning agents. This focus on benchmark improvement can stall real-world progress in RL as findings may not transfer from simulation to the real-world. Thus the deployment of RL agents needs to focus on generalizable, safe, and interpretable RL agents, none of which are the focus of our work.
>
> > Weakness: While the paper reimplements prior algorithms for consistency, there is discrepancy in replication (e.g., PCGrad results are not reproducible, Table 2)
>
> We thank the reviewer for their suggestions on how to rectify our PCGrad results. While it would be great to replicate the results of the PCGrad paper directly, we have verified our implementation across both the official Tensorflow implementation, and a fairly recent work that also implements PCGrad [1]. Unfortunately, it is unclear to us where our issue with results comes from. After validating our implementation we attempted several different hyperparameter settings to attempt to replicate the original work though we were unable to do so. We believe that our contribution is still meaningful as our goal was to focus on the discrepancy between Meta-World versions, while not making any claims about SOTA performance. We will add details of our hyperparameter sweeps and broader experimentation with PCGrad in the appendix of the final version, to assist future researchers who may want to also experiment with this method.
>
> [1] Multi-Task Reinforcement Learning with Context-based Representations. Shagun Sodhani, Amy Zhang, Joelle Pineau. ICML 2021

---

> > ### Author Response · Authors · 2025-08-01
> > **XYZ Base file for Franka Arm (Part 1)**
> >
> > > Section 5.2 (Future Directions), Section 6 (Conclusion) Weakness: Although the paper briefly mentions future directions involving cross-embodiment
> >
> > ## To begin to add support for different arms, we demonstrate the integration of a Franka Emika Panda arm in the next comments.
> >
> > ```
> > <mujocoinclude>
> > <camera pos="0 0.5 1.5" name="topview" />
> > <camera name="corner" mode="fixed" pos="-1.1 -0.4 0.6" xyaxes="-1 1 0 -0.2 -0.2 -1"/>
> > <camera name="corner2" fovy="60" mode="fixed" pos="1.3 -0.2 1.1" euler="3.9 2.3 0.6"/>
> > <camera name="corner3" fovy="45" mode="fixed" pos="0.9 0 1.5" euler="3.5 2.7 1"/>
> > <body name="base" childclass="xyz_base" pos="0 0 0">
> > <site name="basesite" pos="0 0 0" size="0.01" />
> > <inertial pos="0 0 0" mass="0" diaginertia="0 0 0" />
> > <body name="controller_box" pos="0 0 0">
> > <inertial pos="-0.325 0 -0.38" mass="46.64" diaginertia="1.71363 1.27988 0.809981" />
> > <geom size="0.11 0.2 0.265" pos="-0.325 0 -0.38" type="box" rgba="0.2 0.2 0.2 1"/>
> > </body>
> > <body name="pedestal_feet" pos="0 0 0">
> > <inertial pos="-0.1225 0 -0.758" mass="167.09" diaginertia="8.16095 9.59375 15.0785" />
> > <geom size="0.385 0.35 0.155" pos="-0.1225 0 -0.758" type="box" rgba="0.2 0.2 0.2 1" contype="0" conaffinity="0"/>
> > </body>
> > <body name="torso" pos="0 0 0">
> > <inertial pos="0 0 0" mass="0.0001" diaginertia="1e-08 1e-08 1e-08" />
> > <geom size="0.05 0.05 0.05" type="box" contype="0" conaffinity="0" group="1" rgba="0.2 0.2 0.2 1" />
> > </body>
> > <body name="pedestal" pos="0 0 0">
> > <inertial pos="0 0 0" quat="0.659267 -0.259505 -0.260945 0.655692" mass="60.864" diaginertia="6.0869 5.81635 4.20915" />
> > <geom pos="0.26 0.345 -0.91488" quat="0.5 0.5 -0.5 -0.5" type="mesh" contype="0" conaffinity="0" group="1" rgba="0.2 0.2 0.2 1" mesh="pedestal" />
> > <geom size="0.18 0.31" pos="-0.02 0 -0.29" type="cylinder" rgba="0.2 0.2 0.2 0" />
> > </body>
> > <body name="right_arm_base_link" pos="0 0 0">
> > <inertial pos="-0.0006241 -2.8025e-05 0.065404" quat="-0.209285 0.674441 0.227335 0.670558" mass="2.0687" diaginertia="0.00740351 0.00681776 0.00672942" />
> > <geom type="mesh" contype="0" conaffinity="0" group="1" rgba="0.5 0.1 0.1 1" mesh="base" />
> > <geom size="0.08 0.12" pos="0 0 0.12" type="cylinder" rgba="0.5 0.1 0.1 0" />
> > <body name="right_l0" pos="0 0 0.08">
> > <inertial pos="0.024366 0.010969 0.14363" quat="0.894823 0.00899958 -0.170275 0.412573" mass="5.3213" diaginertia="0.0651588 0.0510944 0.0186218" />
> > <joint name="right_j0" pos="0 0 0" axis="0 0 1" limited="true" range="-3.0503 3.0503" damping="10"/>
> > <geom type="mesh" contype="0" conaffinity="0" group="1" rgba="0.5 0.1 0.1 1" mesh="l0" />
> > <body name="head" pos="0 0 0.2965">
> > <inertial pos="0.0053207 -2.6549e-05 0.1021" quat="0.999993 7.08405e-05 -0.00359857 -0.000626247" mass="1.5795" diaginertia="0.0118334 0.00827089 0.00496574" />
> > <geom type="mesh" contype="0" conaffinity="0" group="1" rgba="0.5 0.1 0.1 1" mesh="head" />
> > <!-- <geom size="0.18" pos="0 0 0.08" rgba="0.5 0.1 0.1 0" /> -->
> > <body name="screen" pos="0.03 0 0.105" quat="0.5 0.5 0.5 0.5">
> > <inertial pos="0 0 0" mass="0.0001" diaginertia="1e-08 1e-08 1e-08" />
> > <geom size="0.12 0.07 0.001" type="box" contype="0" conaffinity="0" group="1" rgba="0.2 0.2 0.2 0" />
> > </body>
> > <body name="head_camera" pos="0.0228027 0 0.216572" quat="0.342813 -0.618449 0.618449 -0.342813">
> > <inertial pos="0.0228027 0 0.216572" quat="0.342813 -0.618449 0.618449 -0.342813" mass="0" diaginertia="0 0 0" />
> > <site name="headsite" pos="0 0 0" size="0.01" />
> > </body>
> > </body>
> > <body name="right_torso_itb" pos="-0.055 0 0.22" quat="0.707107 0 -0.707107 0">
> > <inertial pos="0 0 0" mass="0.0001" diaginertia="1e-08 1e-08 1e-08" />
> > </body>
> > <body name="right_l1" pos="0.081 0.05 0.237" quat="0.5 -0.5 0.5 0.5">
> > <inertial pos="-0.0030849 -0.026811 0.092521" quat="0.424888 0.891987 0.132364 -0.0794296" mass="4.505" diaginertia="0.0224339 0.0221624 0.0097097" />
> > <joint name="right_j1" pos="0 0 0" axis="0 0 1" limited="true" range="-3.8 -0.5" damping="10"/>
> > <geom type="mesh" contype="0" conaffinity="0" group="1" rgba="0.5 0.1 0.1 1" mesh="l1" />
> > <body name="right_l2" pos="0 -0.14 0.1425" quat="0.707107 0.707107 0 0">
> > <inertial pos="-0.00016044 -0.014967 0.13582" quat="0.707831 -0.0524761 0.0516007 0.702537" mass="1.745" diaginertia="0.0257928 0.025506 0.00292515" />
> > <joint name="right_j2" pos="0 0 0" axis="0 0 1" limited="true" range="-3.0426 3.0426" damping="10"/>
> > <geom type="mesh" contype="0" conaffinity="0" group="1" rgba="0.5 0.1 0.1 1" mesh="l2" />
> > <geom size="0.06 0.17" pos="0 0 0.08" type="cylinder" rgba="0.5 0.1 0.1 0" />
> > <body name="right_l3" pos="0 -0.042 0.26" quat="0.707107 -0.707107 0 0">
> > <site name="armsite" pos="0 0 0" size="0.01" />
> > <inertial pos="-0.0048135 -0.0281 -0.084154" quat="0.902999 0.385391 -0.0880901 0.168247" mass="2.5097" diaginertia="0.0102404 0.0096997 0.00369622" />
> > <joint name="right_j3" pos="0 0 0" axis="0 0 1" limited="true" range="-3.0439 3.0439" damping="10"/>
> > <geom type="mesh" contype="0" conaffinity="0" group="1" rgba="0.5 0.1 0.1 1" mesh="l3" />
> > ```

---

> > > ### Author Response · Authors · 2025-08-01
> > > **XYZ Base Part 2**
> > >
> > > ```
> > > <!-- <geom size="0.06" pos="0 -0.01 -0.12" rgba="0.5 0.1 0.1 0" /> -->
> > > <body name="right_l4" pos="0 -0.125 -0.1265" quat="0.707107 0.707107 0 0">
> > > <inertial pos="-0.0018844 0.0069001 0.1341" quat="0.803612 0.031257 -0.0298334 0.593582" mass="1.1136" diaginertia="0.0136549 0.0135493 0.00127353" />
> > > <joint name="right_j4" pos="0 0 0" axis="0 0 1" limited="true" range="-2.9761 2.9761" damping="10" />
> > > <geom type="mesh" contype="0" conaffinity="0" group="1" rgba="0.5 0.1 0.1 1" mesh="l4" />
> > > <geom size="0.045 0.15" pos="0 0 0.11" type="cylinder" rgba="0.5 0.1 0.1 0" />
> > > <body name="right_arm_itb" pos="-0.055 0 0.075" quat="0.707107 0 -0.707107 0">
> > > <inertial pos="0 0 0" mass="0.0001" diaginertia="1e-08 1e-08 1e-08" />
> > > </body>
> > > <body name="right_l5" pos="0 0.031 0.275" quat="0.707107 -0.707107 0 0">
> > > <inertial pos="0.0061133 -0.023697 0.076416" quat="0.404076 0.9135 0.0473125 0.00158335" mass="1.5625" diaginertia="0.00474131 0.00422857 0.00190672" />
> > > <joint name="right_j5" pos="0 0 0" axis="0 0 1" limited="true" range="-2.9761 2.9761" damping="10"/>
> > > <geom type="mesh" contype="0" conaffinity="0" group="1" rgba="0.5 0.1 0.1 1" mesh="l5" />
> > > <body name="right_hand_camera" pos="0.039552 -0.033 0.0695" quat="0.707107 0 0.707107 0">
> > > <inertial pos="0.039552 -0.033 0.0695" quat="0.707107 0 0.707107 0" mass="0" diaginertia="0 0 0" />
> > > </body>
> > > <body name="right_wrist" pos="0 0 0.10541" quat="0.707107 0.707107 0 0">
> > > <inertial pos="0 0 0.10541" quat="0.707107 0.707107 0 0" mass="0" diaginertia="0 0 0" />
> > > </body>
> > > <body name="right_l6" pos="0 -0.11 0.1053" quat="0.0616248 0.06163 -0.704416 0.704416">
> > > <inertial pos="-8.0726e-06 0.0085838 -0.0049566" quat="0.479044 0.515636 -0.513069 0.491322" mass="0.3292" diaginertia="0.000360258 0.000311068 0.000214974" />
> > > <joint name="right_j6" pos="0 0 0" axis="0 0 1" limited="true" range="-4.7124 4.7124" damping="10"/>
> > > <geom type="mesh" contype="4" conaffinity="2" group="1" rgba="0.5 0.1 0.1 1" mesh="l6" />
> > > <geom size="0.055 0.025" pos="0 0.015 -0.01" type="cylinder" rgba="0.5 0.1 0.1 0" />
> > > <body name="right_hand" pos="0 0 0.0245" quat="0.707107 0 0 0.707107">
> > > <inertial pos="1e-08 1e-08 1e-08" quat="0.820473 0.339851 -0.17592 0.424708" mass="1e-08" diaginertia="1e-08 1e-08 1e-08" />
> > > <geom type="mesh" contype="1" conaffinity="1" group="1" rgba="0.5 0.1 0.1 1" pos= "0 0 0.03" mesh="eGripperBase"/>
> > > <geom size="0.035 0.014" pos="0 0 0.015" type="cylinder" rgba="0 0 0 1"/>
> > > <body name="hand" pos="0 0 0.12" quat="-1 0 1 0">
> > > <camera name="behindGripper" mode="track" pos="0 0 -0.5" quat="0 1 0 0" fovy="60" />
> > > <camera name="gripperPOV" mode="track" pos="0.04 -0.06 0" quat="-1 -1.3 0 0" fovy="90" />
> > > <site name="endEffector" pos="0.04 0 0" size="0.01" rgba='1 1 1 0' />
> > > <geom name="rail" type="box" pos="-0.05 0 0" density="7850" size="0.005 0.055 0.005"  rgba="0.5 0.5 0.5 1.0" condim="3" friction="2 0.1 0.002"   />
> > > <body name="rightclaw" pos="0 -0.05 0" >
> > > <geom class="base_col" name="rightclaw_it" condim="4" margin="0.001" type="box" user="0" pos="0 0 0" size="0.045 0.003 0.015"  rgba="1 1 1 1.0"   />
> > > <joint name="r_close" pos="0 0 0" axis="0 1 0" range= "0 0.04" armature="100" damping="1000" limited="true"  type="slide"/>
> > > <site name="rightEndEffector" pos="0.045 0 0" size="0.01" rgba="1.0 0.0 0.0 1.0"/>
> > > <body name="rightpad" pos ="0 .003 0" >
> > > <geom name="rightpad_geom" condim="4" margin="0.001" type="box" user="0" pos="0 0 0" size="0.045 0.003 0.015" rgba="1 1 1 1.0" solimp="0.95 0.99 0.01" solref="0.01 1" friction="2 0.1 0.002" contype="1" conaffinity="1" mass="1"/>
> > > </body>
> > > </body>
> > > <body name="leftclaw" pos="0 0.05 0">
> > > <geom class="base_col" name="leftclaw_it" condim="4" margin="0.001" type="box" user="0" pos="0 0 0" size="0.045 0.003 0.015"  rgba="0 1 1 1.0"  />
> > > <joint name="l_close" pos="0 0 0" axis="0 1 0" range= "-0.03 0" armature="100" damping="1000" limited="true"  type="slide"/>
> > > <site name="leftEndEffector" pos="0.045 0 0" size="0.01" rgba="1.0 0.0 0.0 1.0"/>
> > > <body name="leftpad" pos ="0 -.003 0" >
> > > <geom name="leftpad_geom" condim="4" margin="0.001" type="box" user="0" pos="0 0 0" size="0.045 0.003 0.015" rgba="0 1 1 1.0" solimp="0.95 0.99 0.01" solref="0.01 1" friction="2 0.1 0.002"  contype="1" conaffinity="1" />
> > > </body>
> > > </body>
> > > </body>
> > > </body>
> > > </body>
> > > </body>
> > > <body name="right_l4_2" pos="0 0 0">
> > > <inertial pos="1e-08 1e-08 1e-08" quat="0.820473 0.339851 -0.17592 0.424708" mass="1e-08" diaginertia="1e-08 1e-08 1e-08" />
> > > </body>
> > > </body>
> > > </body>
> > > <body name="right_l2_2" pos="0 0 0">
> > > <inertial pos="1e-08 1e-08 1e-08" quat="0.820473 0.339851 -0.17592 0.424708" mass="1e-08" diaginertia="1e-08 1e-08 1e-08" />
> > > </body>
> > > </body>
> > > <body name="right_l1_2" pos="0 0 0">
> > > <inertial pos="1e-08 1e-08 1e-08" quat="0.820473 0.339851 -0.17592 0.424708" mass="1e-08" diaginertia="1e-08 1e-08 1e-08" />
> > > <geom size="0.07 0.07" pos="0 0 0.035" type="cylinder" rgba="0.2 0.2 0.2 0"/>
> > > </body></body></body></body></body>
> > > <body mocap="true" name="mocap" pos="0 0 0">
> > >
> > > ```

---

> > > > ### Author Response · Authors · 2025-08-01
> > > > **XYZ Base Part 3**
> > > >
> > > > ```
> > > > <geom conaffinity="0" contype="0" pos="0 0 0" rgba="0.0 0.5 0.5 0" size="0.01" type="sphere"></geom>
> > > > <site name="mocap" pos="0 0 0" rgba="0.0 0.5 0.5 0" size="0.01" type="sphere"></site>
> > > > </body>
> > > > </mujocoinclude>
> > > > ```

---

> > > > > ### Author Response · Authors · 2025-08-01
> > > > > **XYZ Dependencies**
> > > > >
> > > > > ```
> > > > > <mujocoinclude>
> > > > > <compiler angle="radian" inertiafromgeom="auto" inertiagrouprange="4 5"/>
> > > > > <asset>
> > > > > <material name="xyz_col" rgba="0.3 0.3 1.0 0.5" shininess="0" specular="0.5"/>
> > > > > <mesh file="../objects/meshes/xyz_base/base.stl" name="base"/>
> > > > > <mesh file="../objects/meshes/xyz_base/eGripperBase.stl" name="eGripperBase"/>
> > > > > <mesh file="../objects/meshes/xyz_base/head.stl" name="head"/>
> > > > > <mesh file="../objects/meshes/xyz_base/l0.stl" name="l0"/>
> > > > > <mesh file="../objects/meshes/xyz_base/l1.stl" name="l1"/>
> > > > > <mesh file="../objects/meshes/xyz_base/l2.stl" name="l2"/>
> > > > > <mesh file="../objects/meshes/xyz_base/l3.stl" name="l3"/>
> > > > > <mesh file="../objects/meshes/xyz_base/l4.stl" name="l4"/>
> > > > > <mesh file="../objects/meshes/xyz_base/l5.stl" name="l5"/>
> > > > > <mesh file="../objects/meshes/xyz_base/l6.stl" name="l6"/>
> > > > > <mesh file="../objects/meshes/xyz_base/pedestal.stl" name="pedestal"/>
> > > > > <material class="panda" name="white" rgba="1 1 1 1"/>
> > > > > <material class="panda" name="off_white" rgba="0.901961 0.921569 0.929412 1"/>
> > > > > <material class="panda" name="black" rgba="0.25 0.25 0.25 1"/>
> > > > > <material class="panda" name="green" rgba="0 1 0 1"/>
> > > > > <material class="panda" name="light_blue" rgba="0.039216 0.541176 0.780392 1"/>
> > > > > <mesh name="link0_c" file="link0.stl"/>
> > > > > <mesh name="link1_c" file="link1.stl"/>
> > > > > <mesh name="link2_c" file="link2.stl"/>
> > > > > <mesh name="link3_c" file="link3.stl"/>
> > > > > <mesh name="link4_c" file="link4.stl"/>
> > > > > <mesh name="link5_c0" file="link5_collision_0.obj"/>
> > > > > <mesh name="link5_c1" file="link5_collision_1.obj"/>
> > > > > <mesh name="link5_c2" file="link5_collision_2.obj"/>
> > > > > <mesh name="link6_c" file="link6.stl"/>
> > > > > <mesh name="link7_c" file="link7.stl"/>
> > > > > <mesh name="hand_c" file="hand.stl"/>
> > > > > <mesh file="link0_0.obj"/>
> > > > > <mesh file="link0_1.obj"/>
> > > > > <mesh file="link0_2.obj"/>
> > > > > <mesh file="link0_3.obj"/>
> > > > > <mesh file="link0_4.obj"/>
> > > > > <mesh file="link0_5.obj"/>
> > > > > <mesh file="link0_7.obj"/>
> > > > > <mesh file="link0_8.obj"/>
> > > > > <mesh file="link0_9.obj"/>
> > > > > <mesh file="link0_10.obj"/>
> > > > > <mesh file="link0_11.obj"/>
> > > > > <mesh file="link1.obj"/>
> > > > > <mesh file="link2.obj"/>
> > > > > <mesh file="link3_0.obj"/>
> > > > > <mesh file="link3_1.obj"/>
> > > > > <mesh file="link3_2.obj"/>
> > > > > <mesh file="link3_3.obj"/>
> > > > > <mesh file="link4_0.obj"/>
> > > > > <mesh file="link4_1.obj"/>
> > > > > <mesh file="link4_2.obj"/>
> > > > > <mesh file="link4_3.obj"/>
> > > > > <mesh file="link5_0.obj"/>
> > > > > <mesh file="link5_1.obj"/>
> > > > > <mesh file="link5_2.obj"/>
> > > > > <mesh file="link6_0.obj"/>
> > > > > <mesh file="link6_1.obj"/>
> > > > > <mesh file="link6_2.obj"/>
> > > > > <mesh file="link6_3.obj"/>
> > > > > <mesh file="link6_4.obj"/>
> > > > > <mesh file="link6_5.obj"/>
> > > > > <mesh file="link6_6.obj"/>
> > > > > <mesh file="link6_7.obj"/>
> > > > > <mesh file="link6_8.obj"/>
> > > > > <mesh file="link6_9.obj"/>
> > > > > <mesh file="link6_10.obj"/>
> > > > > <mesh file="link6_11.obj"/>
> > > > > <mesh file="link6_12.obj"/>
> > > > > <mesh file="link6_13.obj"/>
> > > > > <mesh file="link6_14.obj"/>
> > > > > <mesh file="link6_15.obj"/>
> > > > > <mesh file="link6_16.obj"/>
> > > > > <mesh file="link7_0.obj"/>
> > > > > <mesh file="link7_1.obj"/>
> > > > > <mesh file="link7_2.obj"/>
> > > > > <mesh file="link7_3.obj"/>
> > > > > <mesh file="link7_4.obj"/>
> > > > > <mesh file="link7_5.obj"/>
> > > > > <mesh file="link7_6.obj"/>
> > > > > <mesh file="link7_7.obj"/>
> > > > > <mesh file="hand_0.obj"/>
> > > > > <mesh file="hand_1.obj"/>
> > > > > <mesh file="hand_2.obj"/>
> > > > > <mesh file="hand_3.obj"/>
> > > > > <mesh file="hand_4.obj"/>
> > > > > <mesh file="finger_0.obj"/>
> > > > > <mesh file="finger_1.obj"/>
> > > > > </asset>
> > > > > <default>
> > > > > <default class="xyz_base">
> > > > > <joint armature="0.001" damping="2" limited="true"/>
> > > > > <geom conaffinity="0" contype="0" group="1" type="mesh"/>
> > > > > <position ctrllimited="true" ctrlrange="0 1.57"/>
> > > > > <default class="base_viz">
> > > > > <geom conaffinity="0" condim="4" contype="0" group="1" margin="0.001" solimp=".8 .9 .01" solref=".02 1" type="mesh"/>
> > > > > </default>
> > > > > <default class="base_col">
> > > > > <geom conaffinity="1" condim="4" contype="1" group="4" margin="0.001" material="xyz_col" solimp=".8 .9 .01" solref=".02 1"/>
> > > > > </default>
> > > > > </default>
> > > > > <default class="panda">
> > > > > <material specular="0.5" shininess="0.25"/>
> > > > > <joint armature="0.1" damping="1" axis="0 0 1" range="-2.8973 2.8973"/>
> > > > > <general dyntype="none" biastype="affine" ctrlrange="-2.8973 2.8973" forcerange="-87 87"/>
> > > > > <default class="finger">
> > > > > <joint axis="0 1 0" type="slide" range="0 0.04"/>
> > > > > </default>
> > > > > <default class="visual">
> > > > > <geom type="mesh" contype="0" conaffinity="0" group="2"/>
> > > > > </default>
> > > > > <default class="collision">
> > > > > <geom type="mesh" group="3"/>
> > > > > <default class="fingertip_pad_collision_1">
> > > > > <geom type="box" size="0.0085 0.004 0.0085" pos="0 0.0055 0.0445"/>
> > > > > </default>
> > > > > <default class="fingertip_pad_collision_2">
> > > > > <geom type="box" size="0.003 0.002 0.003" pos="0.0055 0.002 0.05"/>
> > > > > </default>
> > > > > <default class="fingertip_pad_collision_3">
> > > > > <geom type="box" size="0.003 0.002 0.003" pos="-0.0055 0.002 0.05"/>
> > > > > </default>
> > > > > <default class="fingertip_pad_collision_4">
> > > > > <geom type="box" size="0.003 0.002 0.0035" pos="0.0055 0.002 0.0395"/>
> > > > > </default>
> > > > > <default class="fingertip_pad_collision_5">
> > > > > <geom type="box" size="0.003 0.002 0.0035" pos="-0.0055 0.002 0.0395"/>
> > > > > </default>
> > > > > </default>
> > > > > </default>
> > > > > </default>
> > > > > </mujocoinclude>
> > > > > ```

---

> > > > > > ### Author Response · Authors · 2025-08-06
> > > > > >
> > > > > > Hi reviewer xUrJ, we hope that we have satisfactorily answered all your questions. As we near the end of the discussion period, are there any remaining concerns that prevent you from increasing your score from a borderline reject? If so, we'd be happy to address and/or discuss them!

---

### Official Review · Reviewer_YJtM · 2025-06-19

**Rating:** 5
**Confidence:** 3

**Summary:**

This paper proposes Meta-World+, an enhanced version of the widely-used Meta-World benchmark, for multi-task and meta-reinforcement learning algorithms and agents. The goal of the benchmark is to learn diverse skills simultaneously. Overall, compared to the old version, the updates improve the comparison fairness and also introduce several new components like new task sets and reward functions. In more specific, by noticing that the past comparisons based on old-version reward functions can introduce inconsistency,  the authors provide guidelines for the benchmark designs in a case-by-case manner. Two new task sets are provided and provide the flexibility for users to customize task sets on their own. Finally, the library is updated to be compatible with Gymnasium API and Mujoco Python bindings, which have been released recently. In addition, the process for the code implementations has been greatly simplified and is more user-friendly.

**Additional Feedback:**

In summary, this new benchmark provides sufficient upgrades compared to the original one. Given the great impact of the dataset and benchmark, I tend to vote for weakly acceptance.

**Dataset Code Accessibility:**

Yes

**Dataset Code Comments:**

The author provided the source codes with detailed instructions. In addition, the paper contains necessary details and information about this new update.

**Ethical Considerations:**

No, there are no or only very minor ethics concerns

**Final Justification:**

My questions have been addressed satisfactorily. I raise the score to 'accept'.

**Limitations Weaknesses:**

1.	Although the introduction mentions the key differences from the original version of Meta-World, the main body seems to more focus on Meta-World, and the difference or changes are not that clear. This part should be elaborated more. After reading the paper, the major improvement over the old version seems to be more on the implementation side. Thus, the improvement seems to be slightly incremental, given the main body of the old Meta-World.

2.	For the comparison experiments, the benchmark algorithms might not be sufficient. Many of them were proposed several years ago. For example, PCGrad was proposed in 2020 and MAML in 2017. It might be better to test over more recent approaches, and then make the conclusion with more comprehensive.

3.	The authors only provide insights rather than a systematic approach into dealing with the empirical inconsistency when using old-version reward functions.

**Strengths Contributions:**

1.	Meta-World has been a dominant benchmark for multi-task and meta-RL algorithms. This paper identifies the inconsistency issue in previous version via extensive experiments. This could serve as good insights for fair algorithmic comparisons. The codes have been also upgraded and simplified.  This will greatly enhance the user experiences and further broaden the impact of this open-source datasets.

2.	New tasks are provided and provider users with better flexibility to craft their test setups. This could be very important because MTL or meta-RL often comes with different setups, which could rely on changes or modifications to the original tasks.

3.	The insights on the fair algorithm comparisons like running benchmark methods rather than just copying previous numbers are important and thought-invoking.

---

> ### Author Rebuttal · Authors · 2025-07-29
>
> We thank reviewer YJtM for their time in reviewing our work.
>
> > Although the introduction mentions the key differences from the original version of Meta-World
>
> Before our work there was a conflation of Meta-World versions that undermined fair algorithm comparisons in multi-task and meta-reinforcement learning. Our goal with Meta-World+ is to enable scientific rigor that enables cumulative research progress and reduces validation barriers. While this may seem like an incremental contribution, we argue that this is actually a large contribution. Our work has highlighted the fact that research using Meta-World, especially in multi-task RL, has stagnated due to different versions of Meta-World existing. This can be seen in the raw results table, or the figures from the manuscript, where MOORE, Soft Modularization, and PaCo all perform relatively similarly. This fact is not apparent in the publications for recent methods, which under-report the performance of Soft Modularization. We also highlight the fact that recent meta-RL approaches, such as Amago-2, do not generalize well to the testing tasks in Meta-World+, showing that progress in meta-RL may have stagnated as well.
>
> > For the comparison experiments, the benchmark algorithms might not be sufficient.
>
> Our multi-task results cover a wide selection of relevant baselines from 2020 to 2024, while our meta-RL results were a bit outdated. We have rectified this fact by running a recent meta-RL algorithm, Amago-2, with Meta-World+. These results can be found in the results table (rebuttal to reviewer KifD), and will be added to the camera ready manuscript.
>
> > The authors only provide insights rather than a systematic approach into dealing with the empirical inconsistency when using old-version reward functions.
>
> We respectfully disagree with the characterization that we only provide insights. Our work addresses the empirical inconsistency problem through a concrete, systematic approach with three key components: (a) Systematic Preservation: Rather than creating yet another reward version (V3) that would further fragment the field, we systematically preserve both V1 and V2 reward functions through a standardized API, ensuring researchers can explicitly select which version to use, allowing for both backwards-compatibility and consistency in evaluation. (b) Comprehensive Re-benchmarking Protocol: We provide a systematic methodology by re-evaluating major methods (in both the multi-task RL and meta-RL settings) using both reward versions, establishing clear performance baselines for each version combination.(c) Reproducibility Framework: Meta-World+ operationalizes our approach by providing the complete codebase needed to replicate results from either version, giving researchers a concrete tool to address version inconsistencies in their own work.
>
> This constitutes a systematic approach because it: provides a principled solution (preservation over proliferation), offers a replicable methodology (standardized re-benchmarking), and delivers practical tools (Meta-World+ implementation) that other researchers can immediately adopt to handle version inconsistencies in their experiments. Through our improvements to Meta-World+ we have also upgraded from OpenAI Gym & Mujoco-Py to Farama Foundation’s Gymnasium and Google Deepmind’s Mujoco. These upgrades also leave Meta-World+ to be well positioned to leverage GPU acceleration, once MJX supports the needed collision primitives.

---

> > ### Comment · Reviewer_YJtM · 2025-08-04
> >
> > My questions have been satisfactorily addressed. I keep my recommendation.
> >
> > Best,
> > Reviewer

---

> > > ### Author Response · Authors · 2025-08-04
> > >
> > > Thank you for acknowledging our rebuttal, and we are glad we have satisfactorily answered all your questions. Are there any remaining concerns that prevent you from increasing your score from a borderline accept? If so, we'd be happy to address and/or discuss them!

---

> > > > ### Comment · Reviewer_YJtM · 2025-08-05
> > > >
> > > > I thought I had already gave 'accept'. The score is now updated accordingly.
> > > >
> > > > Best,
> > > > Reviewer

---

### Official Review · Reviewer_KifD · 2025-06-30

**Rating:** 5
**Confidence:** 3

**Summary:**

This work addresses the challenges in evaluating multi-task and meta-reinforcement learning agents using Meta-World, where undocumented changes have hindered fair algorithm comparisons. It aims to clarify literature results and leverage past Meta-World versions to offer insights into benchmark design for such tasks. The authors release an open-source Meta-World version that ensures full reproducibility of previous results, improves technical usability, and empowers users to better control task set inclusions.

**Additional Feedback:**

- This paper lacks an in-depth comparison of the Meta-World+ benchmark against other RL benchmarks in a tabular format, both single-task and multi-task RL benchmarks
- The baseline methods are a bit insufficient. The authors should include more baseline methods in the experiment section
- It would be better if the authors could provide detailed results of baseline methods on evaluated tasks, akin to repos like CORL (https://github.com/tinkoff-ai/CORL)
- A minor point, the authors should not include the citation information in the anonymized repository if they choose to hide their identities

**Dataset Code Accessibility:**

Yes

**Dataset Code Comments:**

Yes, the code is available

**Ethical Considerations:**

No, there are no or only very minor ethics concerns

**Final Justification:**

I am satisfied with the rebuttal from the authors and hence raise my score from 4 to 5

**Limitations Weaknesses:**

The only concern I have with this submission is that this paper seems to be incremental upon Meta-World. I actually do not see that much workload and that significant improvement over the original version of Meta-World.

**Strengths Contributions:**

- The paper adeptly identifies and addresses the critical issue of undocumented changes in Meta-World, which have undermined fair algorithm comparisons in multi-task and meta-reinforcement learning.
- The release of an open-source Meta-World version ensures full reproducibility of historical results, a cornerstone of scientific rigor that enables cumulative research progress and reduces validation barriers.

---

> ### Author Rebuttal · Authors · 2025-07-29
>
> We thank reviewer KifD for their time in reviewing our work. We appreciate the understanding of how valuable this work is, as before this work a conflation of Meta-World versions undermined fair algorithm comparisons in multi-task and meta-reinforcement learning. Our goal with Meta-World+ is to enable scientific rigor that enables cumulative research progress and reduces validation barriers, and we thank the reviewer for recognizing this.
>
> > The only concern I have with this submission is that this paper seems to be incremental upon Meta-World. I actually do not see that much workload and that significant improvement
>
> While the workload may appear to be minimal, there was actually a substantial amount of work that occurred “under the hood.” The majority of our code development work was re-engineering Meta-World to leverage Google Deepmind’s Mujoco and Farama Foundation’s Gymnasium, updating these packages from OpenAI Mujoco-Py and Gym. This allowed us to dig deeper into the differences between versions of Meta-World that we have highlighted in the paper. We would like to emphasize that our work encompasses not just updates to Meta-World but also sorting out relevant results from the literature, extensive benchmarking to uncover incorrect conclusions drawn from the literature, as well as open-source implementations of each algorithm we included in the paper. Each of these contributions alleviate issues with multi-task-RL, meta-RL, and Meta-World that will enable better scientific results with Meta-World going forward. We will add a section in the Appendix to specify in more detail many of the changes that were necessary, to avoid minimizing the effort put in.
>
> > This paper lacks an in-depth comparison of the Meta-World+ benchmark against other RL benchmarks in a tabular format,
>
> Perhaps one of the largest differences between Meta-World+ and other robotic simulation environments (i.e. RLBench, ManiSkill) is that Meta-World has a unified observation & action space. This observation & action space is designed in a way to facilitate work in multi-task and meta-RL, while other simulation environments have different focuses. If the reviewer has any specific suggestions for axes of comparisons for a table of this manner, we would be happy to explore this visualization.
>
> > The baseline methods are a bit insufficient.
>
> We have added the recent meta-RL method Amago-2 to our results (below). We note that Amago-2 reports results on the training set of environments, while our results are on the testing set of environments for both ML10 and ML45. This is the reason for the discrepancy in performance between the reported results of Amago-2 and the results we have here.
>
> > A minor point, the authors should not include the citation information in the anonymized repository
>
> We apologize for the citation issue. We have removed that from the anonymous repo.
>
> > It would be better if the authors could provide detailed results of baseline methods on evaluated tasks
>
> Here are the raw results as suggested (in percentages).  We will also add these to the final version of the paper.
>
>
> ### MT10 V1 Results
> | Algorithm | Result | Std Dev |
> | ------------- |:-------------:|:-------------:|
> | MTMHSAC | 46.88 | 22.69 |
> | Soft Modules | 71.41 | 2.08 |
> | MOORE | 61.44 | 7.10 |
> | PaCo | 26.18 | 7.54 |
> | PCGrad | 70.49 | 2.16 |
>
> ### MT10 V2 Results
> | Algorithm | Result | Std Dev |
> | ------------- |:-------------:|:-------------:|
> | MTMHSAC | 77.26 | 5.64 |
> | Soft Modules | 84.88 | 4.51 |
> | MOORE | 83.42 | 7.93 |
> | PaCo | 72.26 | 7.44 |
> | PCGrad | 85.52 | 1.72 |
>
> ### MT50 V1 Results
> | Algorithm | Result | Std Dev |
> | ------------- |:-------------:|:-------------:|
> | MTMHSAC | 31.92 | 2.67 |
> | Soft Modules | 60.59 | 3.86 |
> | MOORE | 61.772 | 2.66 |
> | PaCo | 18.58 | 15.38 |
> | PCGrad | 45.84 | 5.93 |
>
> ### MT50 V2 Results
> | Algorithm | Result | Std Dev |
> | ------------- |:-------------:|:-------------:|
> | MTMHSAC | 54.90 | 1.07 |
> | Soft Modules | 65.78 | 1.94 |
> | MOORE | 71.99 | 2.93 |
> | PaCo | 58.88 | 4.64 |
> | PCGrad | 69.49 | 2.67 |
>
> ### ML10 V1 Results
> | Algorithm | Result | Std Dev |
> | ------------- |:-------------:|:-------------:|
> | MAML | 39.58 | 16.11 |
> | MAML (no baseline) | 4.00 | 2.83 |
> | RL2 | 6.53 | 7.74 |
> | Amago-2 | (running, to be updated) | - |
>
> ### ML10 V2 Results
> | Algorithm | Result | Std Dev |
> | ------------- |:-------------:|:-------------:|
> | MAML | 30.1 | 11.45 |
> | MAML (no baseline) | 2.00 | 3.46 |
> | RL2 | 25.7 | 8.92 |
> | Amago-2 | 6 | 4 |
>
> ### ML45 V1 Results
> | Algorithm | Result | Std Dev |
> | ------------- |:-------------:|:-------------:|
> | MAML | 26.67 | 7.67 |
> | MAML (no baseline) | 0 | 0 |
> | RL2 | 9.71 | 12.31 |
> | Amago-2 | (running, to be updated) | - |
>
> ### ML45 V2 Results
> | Algorithm | Result | Std Dev |
> | ------------- |:-------------:|:-------------:|
> | MAML | 32.4 | 14.47 |
> | MAML (no baseline) | 0 | 0 |
> | RL2 | 36.76 | 13.90 |
> | Amago-2 | (running, to be updated) | - |

---

> > ### Comment · Reviewer_KifD · 2025-08-01
> >
> > Thank you for the rebuttal.
> >
> > > in-depth comparison of the Meta-World+ benchmark against other RL benchmarks
> >
> > The authors can consider (a) benchmark API; (b) benchmark environment type (continual RL, generalization, visual RL, meta RL, offline RL, general RL, etc.); (c) number of benchmark environments; (d) single agent or multi-agent, single task or multi-task, etc.
> >
> > > The baseline methods are a bit insufficient. It would be better if the authors could provide detailed results of baseline methods on evaluated tasks
> >
> > Please include the baseline results in the repository page, and include more baseline experiments.
> >
> > It seems that most of my concerns are addressed and hence I am raising my score.

---

### Official Review · Reviewer_7oF3 · 2025-07-03

**Rating:** 5
**Confidence:** 4

**Summary:**

The authors introduce Meta-World+, fully versioned, standardized release of the popular Meta-World benchmark for multi-task and meta-reinforcement learning. The authors show that undocumented tweaks to the reward functions have made past results hard to compare, so they rigorously re-evaluate a range of multi-task and meta-RL approaches, under both the original and revised reward schemes, revealing notable shifts in performance. They also introduce two intermediate-scale task collections and provide tools for creating custom task sets, helping future works trade off between compute cost and experimental scope. Last, they modernize the entire codebase to work seamlessly with the latest Gymnasium API and updated Mujoco bindings.

**Dataset Code Accessibility:**

No

**Dataset Code Comments:**

Mentioned in weaknesses.

**Ethical Considerations:**

No, there are no or only very minor ethics concerns

**Final Justification:**

The authors have answered my concerns and I do not see any blockers preventing acceptance of the work.

**Limitations Weaknesses:**

1. Given the long runtimes of the RL experiments, release of the raw data via ML tracking dashboards like Weights & Biases or CSV files would greatly benefit the community.
2. The fields of meta-RL, multi-task RL, and robotics have advanced considerably since Meta-World first appeared in 2020, with a host of new benchmarks now available. While there are virtues in a paper that seeks to standardize benchmarks and promote good practices Meta-World+ ultimately remains a purely simulated benchmark for the Sawyer-arm setting. Focusing exclusively on one setting risks limiting algorithmic innovation and obscuring whether proposed meta-RL methods truly generalize. In my view, introducing additional benchmarks, particularly for different robotic embodiments or vision-based tasks, would serve the community exponentially better. That said, this observation alone does not warrant rejecting the paper.
3. Custom task-set API, though helpful, is not a particularly significant contribution (does not appear to need much more extra coding).
4. The community has been moving toward GPU-accelerated platforms like Mujoco XLA in recent years. By targeting Gymnasium compatibility rather than integrating with MJX’s GPU support, Meta-World+ could inadvertently limit future research efficiency. Again, not a factor for rejection.

**Strengths Contributions:**

1. By directly comparing v1 vs. v2 reward functions across multiple algorithms, the paper demonstrates that many reported advances may stem from reward-scale artifacts rather than algorithmic improvements. This level of transparency is critical for reproducible benchmarking.
2. Gymnasium interfacing is helpful for usability, and custom task-set API allows for more controlled experiments on specific learning challenges, and flexibility in trading off between diversity and computational costs.
2. The key strength of the paper lies in deriving principles for next-generation benchmarks, consistent reward scaling, compositional task diversity, and importantly, good practices in versioning, an aspect which we, the ML research community, admittedly often overlook. While it remains unclear how effective Meta-World+ is for evaluating the generalization capabilities of modern meta-RL and multi-task RL algorithms (since it does not involve cross-embodiment/vision-based tasks), it nevertheless establishes a valuable precedent that will inform  future benchmark design.

---

> ### Author Rebuttal · Authors · 2025-07-29
>
> We thank reviewer 7oF3 for their insightful comments and feedback to make our work stronger. Our goal with Meta-World+ is to enable scientific rigor that enables cumulative research progress and reduces validation barriers. We thank the reviewer for pushing our work to be even stronger in that aspect.
>
> > Given the long runtimes of the RL experiments, release of the raw data
>
> We agree that the release of raw data would be a benefit to the community, and we will release the raw Weights & Biases logs once this work has been published. We will also explore the addition of our Meta-World data to the OpenRL Benchmark [1].
>
> > The fields of meta-RL, multi-task RL, and robotics have advanced considerably since Meta-World first appeared in 2020, with a host of new benchmarks
>
> While the observation that Meta-World+ is a purely simulated benchmark for the Sawyer Arm setting is correct, we observe that useful insights can still be obtained from this simulated setting for researchers without access to real-world robotics hardware. Examples of this is the work performed in [2][3][4][5][6]. Due to the issues with versioning, the same advancements cannot be had in multi-task or meta-RL which is why we have undertaken this work of clearing up the results in the field. From our experiments, we find that there is little to no difference between recent methods (MOORE, PaCo) compared to early methods (Soft Modularization). In fact, the lack of progress with multi-task RL is due to the versioning issues that we have highlighted.
>
> Further, we would also like to note that, while this paper is focused on Meta-World+, it is not meant to imply that other benchmarks are less valuable. We will add clarification in the final version to highlight that Meta-World+ is complementary to other benchmarks and, indeed, additional benchmarks can aid in advancing the field.
>
> > Custom task-set API, though helpful, is not a particularly significant contribution
>
> In Figure 5 of our submitted manuscript, we outline the environment creation process which shows how we have simplified the usage of Meta-World to use the `gym.make` API. If we were to attempt to create a custom task set with the original Meta-World API, we would have to either modify (a) the example given in Figure 5 (left) to behave as required, or (b) modify the code that is ‘under the hood’ of Meta-World. Through our upgrades, we have removed the need for either of these changes by using the `gym.make` API. This API change makes creating a custom task set very easy:
>
> `envs = gym.make(‘Meta-World/MT-custom’ , env_names=[’env1-v3’, ’env2-v3’,... ,’env-v3 ’] , vector_strategy=’sync’, seed = 42)`
>
> This example shows that our changes to use the gym.make API, while not many lines of code of implementation, is a large improvement over the version in Figure 5 (left). In addition to this user friendliness, this also makes new lines of research more accessible where interested researchers can leverage this custom task set API to ask deeper questions about the effects of task sets in multi-task and meta-RL.
>
> > The community has been moving toward GPU-accelerated platforms
>
> We agree that targeting GPU-accelerated platforms like Mujoco MJX is an important feature for benchmarks to strive for. We would like to clarify that we have upgraded to use Gymnasium and Mujoco which leaves Meta-World+ to be well positioned to leverage GPU acceleration. However, Mujoco MJX does not currently support the collisions for some Mujoco primitives that Meta-World+ uses. Once MJX supports the needed collisions, this would be a useful avenue for future work.
>
>
> [1] Open RL Benchmark: Comprehensive Tracked Experiments for Reinforcement Learning Shengyi Huang, Quentin Gallouédec, et al.
>
> [2] A single goal is all you need: Skills and exploration emerge from contrastive rl without rewards, demonstrations, or subgoals. Grace Liu, Michael Tang, Benjamin Eysenbach. ICLR 2025.
>
> [3] Meta-DT: Offline Meta-RL as Conditional Sequence Modeling with World Model Disentanglement. Zhi Wang, Li Zhang, Wenhao Wu, Yuanheng Zhu, Dongbin Zhao, Chunlin Chen. NeurIPs 2024.
>
> [4] A Generalist Agent. Scott Reed, Konrad Zolna, Emilio Parisotto, Sergio Gomez Colmenarejo, Alexander Novikov, Gabriel Barth-Maron, Mai Gimenez, Yury Sulsky, Jackie Kay, Jost Tobias Springenberg, Tom Eccles, Jake Bruce, Ali Razavi, Ashley Edwards, Nicolas Heess, Yutian Chen, Raia Hadsell, Oriol Vinyals, Mahyar Bordbar, Nando de Freitas. Transactions on Machine Learning Research 2022.
>
> [5] R3M: A Universal Visual Representation for Robot Manipulation. Suraj Nair, Aravind Rajeswaran, Vikash Kumar, Chelsea Finn, Abhinav Gupta. Proceedings of The 6th Conference on Robot Learning.
>
> [6] Video-Language Critic: Transferable Reward Functions for Language-Conditioned Robotics. Minttu Alakuijala, Reginald McLean, Isaac Woungang, Nariman Farsad, Samuel Kaski, Pekka Marttinen, Kai Yuan. Transactions on Machine Learning Resarch 2025.

---

### Note · Authors · 2025-08-14

The authors of this work would like to thank the reviewers for their insightful comments that improved the quality of our work. Our goal with Meta-World+ is to enable scientific rigor that enables cumulative research progress and reduces validation barriers, which the reviewers effectively highlighted and suggested ways to further our contributions.

At the suggestion of multiple reviewers, we have integrated additional meta-RL baselines into our work which will be included in the camera ready submission and our public Github repository. In addition to these meta-RL baselines, we will also include the raw scores of each method in our work in the camera ready submission and the Github repository.

At the suggestion of reviewer xUrJ, we also explore the addition of a new arm in the Meta-World+ framework which would enable further research into multi-task/meta-RL approaches across different robotic morphologies. In addition to this, at the suggestion of reviewer xUrJ we explore the difficulty levels of our MT25 task set using episodic returns where we found that we approximated the difficulty level of MT10 & MT50 in MT25 while reducing the computational cost of the MT50 task set.

In summary, these requested changes have improved the quality of our work and aid us in reaching our desired contributions of this work.

---

### Decision · Program_Chairs · 2025-09-18

**Decision:**

Accept (poster)

**Comment:**

The paper introduces Meta-World+, a standardization of the popular Meta-World benchmark. The authors' central claim is that undocumented changes, especially between reward function versions (v1 vs. v2), have created inconsistencies in the literature, making fair comparisons of reinforcement learning algorithms difficult. Through rigorous re-evaluation, they demonstrate that these inconsistencies have clouded research progress, with some algorithmic advances being mere artifacts of the benchmark version. The paper's primary strength, and its core contribution, is the release of a modernized, open-source benchmark that ensures reproducibility, offers updated library support, and provides tools for custom task creation.

Initial reviews raised valid weaknesses, including the concern that the work was an incremental update rather than a novel contribution. Reviewers also pointed to the limited scope of the benchmark, which remains focused on a single Sawyer robot arm, and noted that the set of baseline algorithms could be more comprehensive. However, the authors addressed these points comprehensively during the rebuttal period. They successfully argued the work's significance by showing how their re-benchmarking revealed potential stagnation in the field, a key scientific insight. They also ran new experiments to include a more recent baseline (Amago-2) , provided raw data tables as requested, and demonstrated a clear path toward integrating new robotic hardware to address the scope limitations.

The authors’ proactive and thorough rebuttal successfully resolved the reviewers' primary concerns, leading to a unanimous decision.